# AMORLIP: Efficient Language-Image Pretraining via Amortization

**Haotian Sun[†], Yitong Li[†], Yuchen Zhuang[†], Niao He[‡], Hanjun Dai[§], Bo Dai[†]**
[†]Georgia Institute of Technology    [‡]Swiss Federal Institute of Technology    [§]Precur.ai
haotian.sun@gatech.edu, bodai@cc.gatech.edu

## Abstract

Contrastive Language-Image Pretraining (CLIP) has demonstrated strong zero-shot performance across diverse downstream text-image tasks. Existing CLIP methods typically optimize a contrastive objective using negative samples drawn from each minibatch. To achieve robust representation learning, these methods require extremely large batch sizes and escalate computational demands to hundreds or even thousands of GPUs. Prior approaches to mitigate this issue often compromise downstream performance, prolong training duration, or face scalability challenges with very large datasets. To overcome these limitations, we propose AMORLIP, an efficient CLIP pretraining framework that amortizes expensive computations involved in contrastive learning through lightweight neural networks, which substantially improves training efficiency and performance. Leveraging insights from a spectral factorization of energy-based models, we introduce novel amortization objectives along with practical techniques to improve training stability. Extensive experiments across 38 downstream tasks demonstrate the superior zero-shot classification and retrieval capabilities of AMORLIP, consistently outperforming standard CLIP baselines with substantial relative improvements of up to 12.24%.

## 1 Introduction

Contrastive language-image pretraining methods, such as CLIP [51, 36] and ALIGN [38], have emerged as powerful paradigms for learning general-purpose vision-language representations from large-scale image-text pairs sourced from the web. By optimizing a contrastive objective, these approaches effectively align representations from image and text modalities within a shared embedding space, thereby facilitating robust zero-shot transfer to diverse downstream tasks, such as image classification and cross-modal retrieval [55, 23, 69, 14].

In practice, training CLIP models typically involves optimizing a ranking-based Noise Contrastive Estimation (NCE) objective [47, 28, 29], where negative pairs are sampled from within each minibatch. This

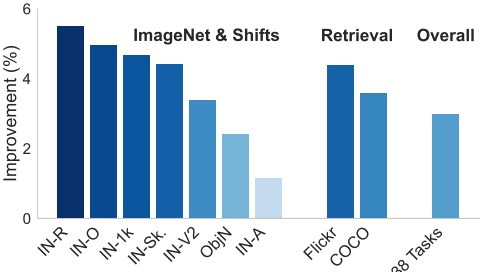

Figure 1: **AMORLIP consistently delivers performance gain over CLIP across various tasks.** The bar plot represents the absolute performance improvements (%) over CLIP [51] in ImageNet classification, retrieval, and overall across the 38 DataComp tasks [23].

minibatch-based negative sampling inherently requires very large batch sizes (e.g., 32K samples or larger [36]) to ensure sufficient diversity among negatives for effective representation learning. A limited number of negative samples can introduce noisy gradient estimates and result in slower convergence and suboptimal downstream task performance. Consequently, CLIP-based models often require significant computational resources, typically involving hundreds or even thousands of GPUs

39th Conference on Neural Information Processing Systems (NeurIPS 2025).

or TPUs [51, 36, 71], thus severely limiting accessibility for practitioners with constrained resources. Moreover, the CLIP objective requires computing similarity scores between all combinations of samples within minibatches before evaluating the loss for each sample pair. This inherent dependency prevents parallel per-sample computations and further hinders training efficiency.

To mitigate these computational barriers, existing works have explored memory-efficient techniques such as unimodal pretraining [72, 36], image masking and rescaling [43, 22, 63, 42], and gradient accumulation methods [11, 36, 12, 16]. Though reducing memory consumption, these methods typically compromise downstream performance or prolong training. Alternatively, recent approaches approximate a larger negative sample set via non-parametric estimation with offline buffers [70, 50, 66, 64]. However, these approaches face scalability challenges, as maintaining buffers comparable to the entire training dataset becomes increasingly impractical when training with billions of samples.

We propose AMORLIP, an efficient CLIP pretraining framework that introduces *amortization* to alleviate the need for large sets of negative samples and significantly enhance training efficiency [2, 58]. We reformulate the CLIP training from an energy-based model perspective and derive an efficient representation for the partition function using spectral factorization. Motivated by this formulation, AMORLIP employs lightweight neural networks to amortize partition functions effectively. We optimize AMORLIP via a two-stage process, alternating updates between lightweight amortization networks and backbone encoders. Through continuous amortization over rolling minibatches, AMORLIP progressively incorporates richer sample information across batches and enables efficient training with minimal overhead. Additionally, we introduce and thoroughly analyze two amortization objectives, accompanied by practical techniques to further enhance training stability and efficiency.

Extensive experiments conducted across 38 downstream tasks demonstrate the robust zero-shot classification and retrieval performance of AMORLIP, achieving substantial relative improvements of up to 12.24% over CLIP. As illustrated in Figure 1, these performance gains are consistent across diverse evaluation settings. Furthermore, comprehensive ablation studies confirm the effectiveness of the proposed representation parameterization for partition functions and validate the impact of our training techniques.

We summarize our main contributions as follows: (1) We propose AMORLIP, an efficient CLIP pretraining framework that amortizes costly computations from CLIP learning via lightweight neural networks, which substantially enhances training efficiency and improves model performance. (2) Leveraging an efficient representation derived from a spectral factorization perspective, AMORLIP effectively approximates partition functions, thereby alleviating the large-batch requirement inherent in CLIP training. (3) Extensive empirical evaluations demonstrate that AMORLIP consistently and significantly outperforms existing CLIP-based methods across diverse downstream tasks.

## 2 Preliminaries

**CLIP as energy-based learning** In this section, we formulate the CLIP objective with energy-based model (EBM) learning. Given a dataset $\mathcal{D} = \left\{ \left( u_I^{(1)}, u_T^{(1)} \right), \ldots, \left( u_I^{(n)}, u_T^{(n)} \right) \right\}$ consisting of paired images $u_I^{(i)}$ and corresponding textual descriptions $u_T^{(i)}$, we pretrain two modality-specific encoders $\psi_I(\cdot)$ and $\psi_T(\cdot)$ to generate representations within a shared embedding space. Let $\psi_I(u_I) \in \mathbb{R}^d$ and $\psi_T(u_T) \in \mathbb{R}^d$ denote the $\ell_2$-normalized embeddings for images and texts. For notation simplicity, we let $l \in \{I, T\}$ represent an arbitrary modality of either image ($I$) or text ($T$). Given modality $l$, we denote the complementary modality as $l' \neq l$. We also use $u_l, u_{l'} \in \{u_I, u_T\}$ to denote the corresponding input. Specifically, both encoders are jointly optimized using the CLIP objective to align representations of matching pairs $\left( u_l^{(i)}, u_{l'}^{(i)} \right)$ while pushing apart representations of non-matching pairs $\left( u_l^{(i)}, u_{l'}^{(j)} \right)$, where $j \neq i$. Generally, the CLIP model fits the conditional distributions $\mathbb{P}(u_l | u_{l'})$. We adopt an energy-based parameterization of these conditional distributions as:

$$\mathbb{P}\left(u_l | u_{l'}\right) = \mathbb{P}\left(u_l\right) \exp\left( \tau \psi_l\left(u_l\right)^\top \psi_{l'}\left(u_{l'}\right) - \log Z_{l'}\left(u_{l'}\right) \right), \tag{1}$$

$$Z_{l'}\left(u_{l'}\right) = \mathbb{E}_{\mathbb{P}(u_l)}\left[ \exp\left( \tau \psi_l\left(u_l\right)^\top \psi_{l'}\left(u_{l'}\right) \right) \right], \ \forall l \in \{I, T\}, \tag{2}$$

where $\mathbb{P}(\cdot)$ denote certain negative sampling distributions from the dataset $\mathcal{D}$, and $\tau$ is a learnable temperature parameter commonly adopted in CLIP-like models [51, 71, 66]; $Z_{l'}(u_{l'})$ is the partition

function to ensure $\mathbb{P}\left(u_l | u_{l'}\right)$ is a valid distribution. The ranking-based NCE objective [47, 28, 29] employed by CLIP can be formulated as follows:

$$\ell_{\text{NCE}} = -\frac{2\tau}{n} \sum_{i=1}^{n} \psi_l\left(u_l^{(i)}\right)^\top \psi_{l'}\left(u_{l'}^{(i)}\right) + \frac{1}{n} \sum_{l \in \{I,T\}} \sum_{i=1}^{n} \log \sum_{u_l^{(j)} \sim P(u_l)} \exp\left(\tau\psi_l\left(u_l^{(j)}\right)^\top \psi_{l'}\left(u_{l'}^{(i)}\right)\right). \quad (3)$$

Existing CLIP implementations [36, 51] typically adopt in-batch negative sampling by contrasting each positive pair against all sample combinations $\left(u_l^{(i)}, u_{l'}^{(j)}\right)$ within each minibatch $\mathcal{B} \subset \mathcal{D}$. Notably, the NCE objective inherently requires large batch sizes to ensure a sufficiently diverse set of negative samples, thereby facilitating effective representation learning.

**$f$-divergence**  Let $p$ and $q$ denote two probability distributions. Given a convex function $f : \mathbb{R}^+ \rightarrow \mathbb{R}$ satisfying $f(1) = 0$ and strict convexity around 1, the $f$-divergence between $q$ and $p$ is defined as:

$$D_f\left(q, p\right) := \mathbb{E}_{p(x)}\left[f\left(\tfrac{q(x)}{p(x)}\right)\right], \quad (4)$$

which measures the discrepancy between the distributions $q$ and $p$ [1]. Many widely used divergences fall under this framework through specific choices of $f(\cdot)$. For instance, the Kullback-Leibler (KL) divergence corresponds to $f(t) = t \log t$, and the Jensen-Shannon (JS) divergence corresponds to $f(t) = \frac{1}{2}\left(t \log t - (t+1) \log \frac{t+1}{2}\right)$.

# 3   AMORLIP: Efficient Amortizations for Partition Functions

In this section, we introduce AMORLIP, an efficient contrastive language-image learning framework that employs lightweight amortization of the partition functions. We briefly outline the proposed framework and defer some detailed derivations and proofs of the preceding statements to Appendix B.

## 3.1   AMORLIP Framework

Despite its widespread adoption, the CLIP objective in Eq. (3) presents two significant challenges: **i) Estimation bias:** $\ell_{\text{NCE}}$ in Eq. (3) is estimated using only the limited number of negative samples from each minibatch, which potentially results in biased gradients, particularly in small-batch scenarios [11, 70]. Consequently, CLIP models employing Eq. (3) require large batch sizes to achieve good contrastive learning performance. **ii) Inter-sample dependency:** The nonlinear `logsumexp` operation in Eq. (3) needs computation over all negative samples prior to evaluating the loss for each individual pair. This inherent inter-sample dependency prevents parallel computations of $\ell_{\text{NCE}}$ and restricts computational efficiency.

To address these challenges, we reformulate the representation learning objective from an EBM perspective. One straightforward approach is Maximum Likelihood Estimation (MLE) on $\mathbb{P}\left(u_{l'} | u_l\right)$:

$$\begin{aligned}
\ell_{\text{MLE}} :=\ & -2\tau\mathbb{E}_{\mathbb{P}(u_l, u_{l'})}\left[\psi_l\left(u_l\right)^\top \psi_{l'}\left(u_{l'}\right)\right] + \sum_{l \in \{I,T\}} \mathbb{E}_{\mathbb{P}(u_l)}\left[\log Z_l\left(u_l\right)\right] \\
\Leftrightarrow\ & -2\tau\mathbb{E}_{\mathbb{P}(u_l, u_{l'})}\left[\psi_l\left(u_l\right)^\top \psi_{l'}\left(u_{l'}\right)\right] + \sum_{l \in \{I,T\}} \mathbb{E}_{\mathbb{P}(u_l)\mathbb{P}(u_{l'})}\left[\tfrac{\exp\left(\tau\psi_l(u_l)^\top \psi_{l'}(u_{l'})\right)}{\texttt{stop\_grad}(Z_l(u_l))}\right],
\end{aligned} \quad (5)$$

where $\mathbb{P}\left(u_l, u_{l'}\right)$ denotes the joint sampling distribution; $\texttt{stop\_grad}\left(\cdot\right)$ stands for the stop-gradient operation; the "$\Leftrightarrow$" indicates gradient equivalence between the two formulations for encoder updates.

While the MLE objective in Eq. (5) effectively models the target conditional distribution $\mathbb{P}(u_{l'} | u_l)$, the computation of the partition function $Z_l\left(u_l\right)$ involves summation over all possible samples $u_l$, making MLE optimization computationally intractable. To make Eq. (5) practical, we aim to construct a learnable representation $\lambda_{\theta_l}\left(u_l\right)$ to approximate $Z_l\left(u_l\right)$ and offload its computation from the MLE optimization. We refer to this strategy as *amortization*, as we amortize the estimation cost of $Z_l\left(u_l\right)$ by separately optimizing $\lambda_{\theta_l}\left(u_l\right)$ over training steps, rather than recomputing it during each forward pass. Concretely, instead of directly optimizing Eq. (5), we decompose the optimization into a modular two-stage training pipeline:

**Stage I (Amortization)**  We first optimize a designed amortization objective $\ell_{\text{amor}}$ to train $\lambda_{\theta_l}\left(u_l\right)$ in approximating $Z_l\left(u_l\right)$, *i.e.*, $\min_{\theta_l} \ell_{\text{amor}}\left(\lambda_{\theta_l}\left(u_l\right)\right)$. In the following section, we explore several design choices for the amortization loss with bias-variance trade-offs.

**Stage II (Representation Learning)** We substitute the optimized $\lambda_{\theta_l}(u_l)$ for $Z_l(u_l)$ in (5):

$$\ell_{\text{MLE}}^{(\text{amor})} := -2\tau \mathbb{E}_{\mathbb{P}(u_l, u_{l'})} \left[ \psi_l(u_l)^\top \psi_{l'}(u_{l'}) \right] + \sum_{l \in \{I, T\}} \mathbb{E}_{\mathbb{P}(u_l)\mathbb{P}(u_{l'})} \left[ \frac{\exp\left(\tau \psi_l(u_l)^\top \psi_{l'}(u_{l'})\right)}{\lambda_{\theta_l}(u_l)} \right]. \quad (6)$$

During training, we alternate optimizations of Stage I and Stage II within each minibatch, with $\mathbb{P}(u_l, u_{l'})$ and $\mathbb{P}(u_l)$ set to the joint and marginal sampling from each minibatch. Since amortization progresses concurrently with representation learning, $\lambda_{\theta_l}(u_l)$ continuously aggregates information across rolling minibatches. This design alleviates the computational burden of repeated partition function calculations executed at each optimization step, thereby mitigating gradient bias during representation learning.

## 3.2 Amortization with Efficient Representation

To establish effective learning objectives for the amortization stage, we first develop an efficient representation of the amortization target $Z_l(u_l)$ from a kernel-based perspective. Recognizing the substantial variance introduced by the learnable temperature scalar $\tau$, we further propose two amortization objectives with the bias-variance trade-off.

**Spectral representation for $Z_l(u_l)$** The EBM parameterization in Eq. (1) naturally leads to a spectral representation of the partition function $Z_l(u_l)$. Specifically, interpreting Eq. (1) as a Gaussian kernel and employing random Fourier features [52, 18, 73], we obtain:

$$\mathbb{P}(u_l | u_{l'}) \propto \mathbb{P}(u_l) \left\langle \phi_\omega(\psi_l(u_l)), \phi_\omega(\psi_{l'}(u_{l'}))^* \right\rangle_{p(\omega)}, \quad (7)$$

where $\omega \sim \mathcal{N}(0, \mathbf{I}_d)$ are the $d$-dimensional random features and the corresponding transform $\phi_\omega(\psi_l(u_l)) := \exp\left(\mathbf{i}\sqrt{\tau}\omega^\top \psi_l(u_l)\right) \exp(\tau/2) \in \mathbb{R}^d$ (Detailed derivation in Appendix B.1).

**Proposition 1.** *The partition function is linearly representable by $\phi_\omega(\psi_l(u_l))$, i.e.,*
$$Z_l(u_l) = \left\langle \phi_\omega(\psi_l(u_l)), v_l \right\rangle_{p(\omega)}.$$

*Proof.* From Eq. (7), there exists a vector $v_l \in \mathbb{R}^d$ such that

$$Z_l(u_l) = \mathbb{E}_{\mathbb{P}(u_{l'})} \left[ \left\langle \phi_\omega(\psi_l(u_l)), \phi_\omega(\psi_{l'}(u_{l'}))^* \right\rangle_{p(\omega)} \right] = \left\langle \phi_\omega(\psi_l(u_l)), \underbrace{\mathbb{E}_{\mathbb{P}(u_{l'})} \left[ \phi_\omega(\psi_{l'}(u_{l'}))^* \right]}_{v_l} \right\rangle_{p(\omega)}.$$
$$\square$$

Motivated by Proposition 1, we introduce lightweight multi-layer perceptrons (MLPs), denoted as $\text{MLP}_{\theta_l}(\psi_l(u_l)) \in \mathbb{R}$, on top of each feature representation $\psi_l(u_l)$ to approximate $v_l^\top \phi_\omega(\psi_l(u_l))$. Additionally, the amortization target $Z_l(u_l)$ can exhibit substantial variance across minibatches due to the learnable temperature scalar $\tau$, which varies considerably during training and may increase up to 100 [51]. To mitigate this numerical instability, we further adopt a log-space parameterization: $\log \lambda_{\theta_l}(u_l) = \text{MLP}_{\theta_l}(\psi_l(u_l))$. Specifically, $\log \lambda_{\theta_l}(u_l)$ learns a scalar within a numerically stable (`float32`-representable) range, and the actual estimate for $Z_l(u_l)$ is subsequently recovered via $\lambda_{\theta_l}(u_l) = \exp\left(\text{MLP}_{\theta_l}(\psi_l(u_l))\right)$. As demonstrated later in Section 4, the proposed small-scale MLPs effectively approximate $Z_l(u_l)$ with negligible computational overhead during training. In the following, we discuss two design choices for the amortization objective.

**Divergence objective for amortization** The learnable function $\lambda_{\theta_l}(u_l)$ amortizes the effect of the partition function $Z_l(u_l)$ and defines an amortized conditional distribution analogous to Eq. (1), *i.e.*, $\mathbb{Q}_{\theta_l}(u_{l'}|u_l) = \mathbb{P}(u_l) \exp\left(\tau \psi_l(u_l)^\top \psi_{l'}(u_{l'}) - \log \lambda_{\theta_l}(u_l)\right)$. We formulate an $f$-divergence objective based on Eq. (4) to minimize the discrepancy between $\mathbb{Q}_{\theta_l}(u_{l'}|u_l)$ and $\mathbb{P}(u_{l'}|u_l)$:

$$\min_{\theta_l} \quad \ell_{\text{amor}, f\text{-div}} := \mathbb{E}_{\mathbb{P}(u_l)} \left[ D_f\left(\mathbb{Q}(u_{l'}|u_l), \mathbb{P}(u_{l'}|u_l)\right) \right]$$
$$= \mathbb{E}_{\mathbb{P}(u_l)\mathbb{P}(u_{l'})} \left[ \exp\left(\tau \psi_l(u_l)^\top \psi_{l'}(u_{l'}) - \log Z_l(u_l)\right) f\left(\frac{Z_l(u_l)}{\lambda_{\theta_l}(u_l)}\right) \right]. \quad (8)$$

The divergence objective introduces an unbiased estimator for $f$-divergence. With a proper choice of $f$, this objective potentially improves numerical stability. For example, we can write the KL divergence with $f(t) = t \log t$:

$$\ell_{\text{amor}, kl\text{-div}} = \mathbb{E}_{\mathbb{P}(u_l)\mathbb{P}(u_{l'})} \left[ \exp\left(\tau \psi_l(u_l)^\top \psi_{l'}(u_{l'}) - \log \lambda_{\theta_l}(u_l)\right) \left(\log \frac{Z_l(u_l)}{\lambda_{\theta_l}(u_l)}\right) \right]. \quad (9)$$

Alternatively, JS divergence is another suitable choice with inherent boundedness that may help further mitigate numerical issues.

---

**Algorithm 1:** AMORLIP Framework

---

1 **Input:** Dataset $\mathcal{D}$; Initial encoders $\psi_l^{(1)}$ and amortization networks $\lambda_{\theta_l}^{(1)}$ for $l \in \{I, T\}$;
Number of epochs: $T$; Number of steps per epoch: $K$; Number of $\lambda_{\theta_l}$ update per batch: $T_\lambda$;
Update interval for $\lambda_{\theta_l}$: $T_{\text{online}}$; Update interval for $\lambda_{\hat{\theta}_l}$: $T_{\text{target}}$.

2 Maintain target networks $\lambda_{\hat{\theta}_l}^{(0)}, \lambda_{\hat{\theta}_l}^{(1)} \leftarrow \lambda_{\theta_l}^{(1)}$

3 **for** $t = 1, \dots, T$ **do**

4    Update $\lambda_{\hat{\theta}_l}^{(t-1)} \leftarrow \lambda_{\hat{\theta}_l}^{(t)}$ and re-initialize $\lambda_{\theta_l}^{(t)}, \lambda_{\hat{\theta}_l}^{(t)}$

5    **for** $k = 1, \dots, K$ **do**

6       Sample $\mathcal{B}_k$ from $\mathcal{D}$ and get $Z_{l,\text{comb}}^{(t)}(u_l)$ via Eq. (12) for each $u_l \in \mathcal{B}_k$ and $l \in \{I, T\}$
         /* Stage I: Amortization */

7       **if** $k \equiv 0 \pmod{T_{online}}$ **then** Optimize $\lambda_{\theta_l}^{(t)}$ via Eq. (10) or Eq. (8) for $T_\lambda$ iterations

8       **if** $k \equiv 0 \pmod{T_{target}}$ **then** Update $\lambda_{\hat{\theta}_l}^{(t)}$ via Eq. (11)
         /* Stage II: Representation Learning */

9       Optimize encoders $\psi_l^{(t)}$ via Eq. (6) for each $u_l \in \mathcal{B}_k$ and $l \in \{I, T\}$.

---

**l2-log objective for amortization**   Observing that each $\lambda_{\theta_l}(u_l)$ is a unary function w.r.t. $u_l$, we can also directly fit $\lambda_{\theta_l}(u_l)$ by matching its log-value at each input point $u_l$:

$$\min_{\theta_l} \ \ell_{\text{amor, l2-log}} := \tfrac{1}{2}\mathbb{E}_{\mathbb{P}(u_l)}\left[\left\|\log \lambda_{\theta_l}(u_l) - \log Z_l(u_l)\right\|^2\right], \tag{10}$$

where $\ell_{\text{amor, l2-log}}$ corresponds to the family of $\ell_{\text{amor, } f\text{-div}}$ described in Eq. (8), with the specific choice of $f(t) = \tfrac{1}{2}(\log t)^2$ (see Appendix B.3 for details).

**Remark (Connection between $\ell_{\text{amor, kl-div}}$ and $\ell_{\text{amor, l2-log}}$):**   Empirically, Eq. (10) introduces a biased estimator for the KL divergence with potentially reduced variance compared to the Monte-Carlo approximation $\ell_{\text{amor, kl-div}}$ in Eq. (9). This variance reduction arises because the optimization in Eq. (10) occurs entirely in log-space $\log \lambda_{\theta_l}(u_l)$, mitigating potential numerical issues caused by the exponential operation $\exp(\cdot)$ presented in Eq. (9). Additionally, $\ell_{\text{amor, l2-log}}$ exhibits relatively low bias, as it closely approximates the KL divergence up to second order under mild conditions (see Appendix B.4 for details). Overall, both $\ell_{\text{amor, kl-div}}$ and $\ell_{\text{amor, l2-log}}$ effectively enhance numerical stability. Thus, we adopt both formulations as design choices for amortization objectives.

### 3.3   Training Techniques for AMORLIP

**Training stability**   The training procedure of AMORLIP involves stochastic approximations at two distinct time scales, alternating optimization between the encoders $\psi_l(\cdot)$ and the partition function estimator $\lambda_{\theta_l}(\cdot)$. Consequently, the optimization frequencies of these components are crucial for stable training. To improve training stability, we introduce a target network $\lambda_{\hat{\theta}_l}(\cdot)$ that slowly updates its parameters towards the online network [48, 26, 9]. Within each training epoch, we update $\lambda_{\hat{\theta}_l}(\cdot)$ every $T_{\text{target}}$ steps using an exponential moving average (EMA) of the online model parameters [44]:

$$\hat{\theta}_l^{(k)} \leftarrow \alpha\hat{\theta}_l^{(k-1)} + (1-\alpha)\theta_l^{(k)}, \tag{11}$$

where $\alpha$ denotes the EMA decay factor, and $k$ represents the current update step. We then substitute $\lambda_{\hat{\theta}_l}(\cdot)$ into Eq. (6) in place of the online network $\lambda_{\theta_l}(u_l)$. Additionally, the rapidly increasing temperature $\tau$ elevates the variance of $Z_l(u_l)$ in the amortization objectives. Denote $\lambda_{\hat{\theta}_l}^{(t)}(\cdot)$ and $Z_l^{(t)}(\cdot)$ as the target amortization network and the partition function at the $t$-th epoch, respectively. We assume the outputs from the target network at the previous epoch, $\lambda_{\hat{\theta}_l}^{(t-1)}(\cdot)$, have a magnitude similar to those of the current online network $\lambda_{\theta_l}^{(t)}(\cdot)$. Thus, we naturally introduce the following weighted combination to replace $Z_l^{(t)}(u_l)$ in Eq. (8) and Eq. (10):

$$Z_{l,\text{comb}}^{(t)}(u_l) = \beta_t\lambda_{\hat{\theta}_l}^{(t-1)}(u_l) + (1-\beta_t)Z_l^{(t)}(u_l), \tag{12}$$

where $\beta_t$ is initialized with a small value when $\tau$ is low and gradually increased up to $\beta_T$ as $\tau$ grows larger. Empirically, we employ a cosine scheduling strategy similar to that in [66]: $\beta_t = \beta_T - 0.5 \cdot \beta_T \left(1 + \cos\frac{\pi t}{T}\right)$. The resulting weighted estimate $Z_{l,\text{comb}}^{(t)}(u_l)$ is thus effectively "flattened" by the target amortization predictions, thus facilitating the optimization of amortization objectives.

Table 1: Performance comparison (%) across (1) top-1 accuracy of zero-shot classification tasks on ImageNet and six distribution shifts, (2) recall@1 of retrieval tasks on Flickr30k and MS-COCO, and (3) overall performance on all 38 DataComp tasks. The results are reported for two training scales. Highest scores are highlighted in **bold**, and second-best scores are underlined. The proposed AMORLIP consistently outperforms baseline methods across all evaluated tasks.

| Tasks ($\rightarrow$) | ImageNet & Dist. Shifts | | | | | | | | Retrieval | | | Avg. 38 Tasks |
|---|---|---|---|---|---|---|---|---|---|---|---|---|
| Method ($\downarrow$) | IN-1k | IN-Sk | IN-V2 | IN-A | IN-O | IN-R | ObjN | *Avg.* | Flickr | COCO | *Avg.* | |
| *ResNet-50 Pretrained on CC3M* | | | | | | | | | | | | |
| CLIP [51] | 16.84 | 10.30 | 13.96 | 3.69 | 21.70 | 20.71 | 11.00 | 14.03 | 25.79 | 13.93 | 19.86 | 21.48 |
| SigLIP [71] | 17.74 | 10.34 | 15.43 | 3.88 | 23.10 | 22.96 | 12.01 | 15.07 | 26.73 | 14.86 | 20.80 | 21.32 |
| SogCLR [70] | 19.91 | 11.91 | 17.90 | 4.27 | 26.05 | 25.69 | 13.51 | 17.03 | 27.51 | 16.57 | 22.04 | 21.47 |
| FastCLIP [66] | 20.58 | 13.03 | 18.09 | 4.15 | 27.10 | 27.22 | 14.04 | 17.74 | 34.31 | 19.80 | 27.06 | 23.46 |
| AMORLIP$_{(f\text{-div})}$ | 21.16 | 13.57 | 18.30 | 4.99 | 27.65 | 28.45 | 14.34 | 18.35 | **35.30** | **19.91** | **27.61** | 24.08 |
| AMORLIP$_{(\text{l2-log})}$ | **21.50** | **14.30** | **19.45** | **5.20** | **28.10** | **29.22** | **14.64** | **18.92** | 35.01 | 19.67 | 27.34 | **24.11** |
| *ViT-B/32 Pretrained on CC12M* | | | | | | | | | | | | |
| CLIP [51] | 25.26 | 15.70 | 21.30 | 4.36 | 29.95 | 33.88 | 12.67 | 20.45 | 34.32 | 17.89 | 26.10 | 27.65 |
| SigLIP [71] | 25.42 | 16.60 | 22.08 | 4.79 | 30.90 | 33.85 | 13.00 | 20.95 | 32.91 | 17.86 | 25.39 | 26.91 |
| SogCLR [70] | 27.59 | 18.28 | 23.01 | 4.83 | 30.45 | 35.43 | 14.14 | 21.96 | 33.01 | 17.33 | 25.17 | 26.97 |
| FastCLIP [66] | 27.74 | 18.33 | 22.51 | 4.72 | 32.45 | 35.72 | 13.77 | 22.18 | 36.64 | 20.78 | 28.71 | 29.00 |
| AMORLIP$_{(f\text{-div})}$ | 29.21 | 19.80 | 24.55 | 5.29 | 34.25 | 39.24 | **15.59** | 23.99 | 37.93 | **22.11** | 30.02 | 29.91 |
| AMORLIP$_{(\text{l2-log})}$ | **29.93** | **20.11** | **24.70** | **5.51** | **34.90** | **39.38** | 15.10 | **24.23** | **38.70** | 21.47 | **30.09** | **30.66** |

**Training efficiency** Unlike the encoders $\psi_l$ requiring optimization at each minibatch, we optimize $\lambda_{\theta_l}$ every $T_{\text{online}}$ encoder optimization steps. Furthermore, the proposed AMORLIP inherently supports efficient multi-GPU training via distributed data parallelism (DDP). Conventional CLIP models [36, 72] require invoking `all_gather(·)` operations *at every step* when optimizing the NCE loss in Eq. (3). In contrast, AMORLIP triggers the gathering operation only during the amortization stage, executed merely $1/T_{\text{online}}$ times as frequently as CLIP. During the contrastive learning stage, AMORLIP computes the amortized partition function using lightweight MLPs independently on each device without calling `all_gather(·)`. Consequently, with a suitably large $T_{\text{online}}$, AMORLIP effectively reduces computational overhead and enhances overall training efficiency. We summarize the proposed AMORLIP in Algorithm 1.

## 4 Evaluation

**Training setups** We compare AMORLIP against widely adopted language-image baselines, including **CLIP** [51], **SigLIP** [71], **SogCLR** [70], and **FastCLIP** [66]. We use the OpenCLIP [36] codebase and original implementations for these models. Following the experimental setups from [66], we pre-train models at two scales: a medium-scale experiment using ResNet-50 [31] trained on Conceptual Captions 3M (CC-3M; [57]) with a batch size of 1024 for 30 epochs, and a large-scale experiment using ViT-B/32 trained on Conceptual Captions 12M (CC-12M; [10]) with a batch size of 2048 for 33 epochs. Due to expired source links, our downloaded datasets contain 2,274,566 samples for CC-3M and 8,059,642 samples for CC-12M. All experiments are conducted using NVIDIA H100 GPUs with 80GB VRAM. Additional training details can be found in Appendix C. Our implementation is available at https://github.com/haotiansun14/AmorLIP.

**AMORLIP implementation** In AMORLIP, we implement $\lambda_{\theta_l}$ using a three-layer MLP for each modality $l \in \{I, T\}$, operating in parallel to the respective text and image encoders. Each MLP takes the corresponding encoder's $d$-dimensional feature as input and outputs a scalar representing the amortized partition function. We control the network's width through a dimension factor $f_d$, setting the intermediate layer dimension as $f_d \cdot d$. Specifically, we choose $f_d = 0.5$ for the medium-scale setting and $f_d = 1.0$ for the large-scale setting. For amortization hyperparameters detailed in Algorithm 1, we set $T_\lambda = 3$ and $T_{\text{target}} = 2$ for both training scales, while $T_{\text{online}}$ is set to 8 for medium-scale and 1 for large-scale experiments. Regarding techniques described in Section 3.3, the EMA factor $\alpha$ is set to 0.999 for medium-scale and 0.92 for large-scale training. The parameter $\beta_T$ is universally fixed at 0.8.

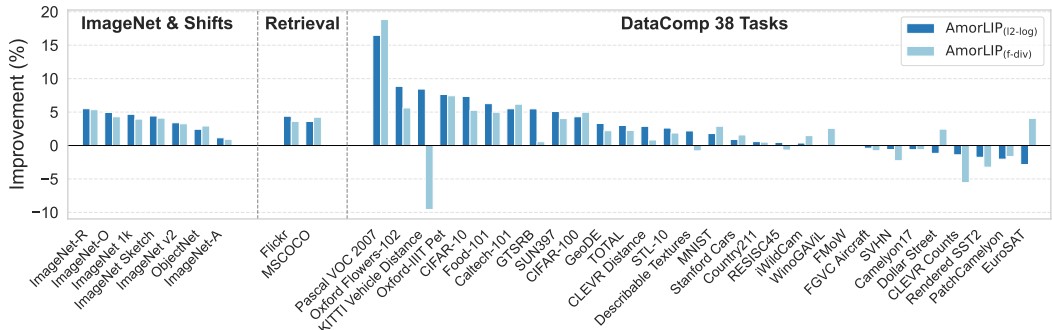

Figure 2: Breakdown of absolute improvement (%) made by AMORLIP over CLIP model on all 38 DataComp Tasks [23] under large scale setting.

**Evaluation metrics**   We evaluate AMORLIP and baseline methods using the DataComp benchmark [23], comprising 38 widely used text-image tasks. Specifically, we report top-1 zero-shot classification accuracy on **ImageNet** (IN-1K; [55]) and six of its distribution-shifted variants: **ImageNet-Sketch** (IN-Sk; [65]), **ImageNet-V2** (IN-V2; [53]), **ImageNet-A** (IN-A; [33]), **ImageNet-O** (IN-O; [33]), **ImageNet-R** (IN-R; [32]), and **ObjectNet** (ObjN; [4]). Additionally, we evaluate retrieval performance via recall@1 on **Flickr30k** (Flickr; [69]) and **MSCOCO** (COCO; [14]). Finally, we report average performance across all 38 **DataComp** tasks (Avg.38). To assess the training efficiency of AMORLIP, we also measure per-step training time and GPU memory (VRAM) usage.

## 4.1   Main Results

Table 1 presents the performance of text-image models across the 38 downstream tasks of DataComp [23]. Consistently, across different encoder architectures and dataset scales, AMORLIP outperforms all baselines in most zero-shot classification and retrieval tasks. Specifically, AMORLIP achieves improvements in top-1 accuracy of up to 4.67% on ImageNet zero-shot classification tasks and up to 4.32% on its distribution-shifted variants. In retrieval tasks, AMORLIP surpasses other methods by an average of 7.75% for the medium-scale experiments and 3.92% for the large-scale experiments. Overall, AMORLIP exhibits substantial relative improvements over CLIP, with 12.24% in the medium-scale setting and 10.89% in the large-scale setting. Additionally, AMORLIP using the l2-log objective demonstrates slightly more consistent performance gains compared to the $f$-div objective, while the $f$-div objective achieves comparable or even superior performance in retrieval tasks. Figure 2 further illustrates that both AMORLIP objectives can achieve up to a 20% absolute improvement on 30 out of the 38 evaluated tasks. These results collectively highlight that the amortization of AMORLIP can effectively enhance multimodal representation pertaining.

## 4.2   Learning Efficiency

**Faster training convergence**   Figure 3 illustrates the evolution of model performance (reflected by classification accuracy) over training epochs. In both evaluated settings, AMORLIP consistently achieves higher final performance than the baseline models. In the medium-scale setting shown in Figure 3a, AMORLIP surpasses the best baseline performance (20.58% by FastCLIP) using only 26 epochs, and achieves convergence around 13.3% faster than all baselines. Further training up to 30 epochs improves the accuracy to 21.50%. In the large-scale setting depicted by Figure 3b, AMORLIP extends this training speed advantage significantly, reaching comparable performance about 10 epochs earlier and equivalently at least 30.3% faster than all baselines. The AMORLIP ultimately achieves a 7.89% relative performance gain over the best-performing baseline. Notably, the efficiency benefit becomes more pronounced in the large-scale scenario, as the increased number of iterations and samples better facilitates amortization optimization. Additionally, AMORLIP initially trails some baselines, such as SogCLR, but begins to outper-

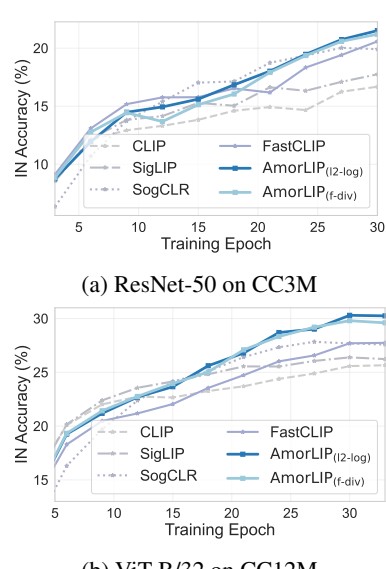

(a) ResNet-50 on CC3M

(b) ViT-B/32 on CC12M

Figure 3: ImageNet classification accuracy (%) of models at two scales.

Table 2: Performance (%) and relative training overhead (%) of AMORLIP at medium-scale under different amortization settings, evaluated on one H100 GPU. The relative overhead is depicted by per-step training time and total VRAM usage (including encoders and amortization), relative to CLIP.

<table>
<tr><td colspan="5">(a) Online update frequency $T_{\text{online}}^{-1}$</td><td colspan="5">(b) Dimension factors $f_d$</td></tr>
<tr><td>$T_{\text{online}}^{-1}$</td><td>IN&Shifts</td><td>Retrieval</td><td>Avg. 38</td><td>$\Delta$Time</td><td>$f_d$</td><td>IN&Shifts</td><td>Retrieval</td><td>Avg. 38</td><td>$\Delta$VRAM</td></tr>
<tr><td>1</td><td>18.28</td><td>27.97</td><td>23.33</td><td>4.47%</td><td>2</td><td>18.78</td><td>28.29</td><td>23.97</td><td>0.60%</td></tr>
<tr><td>1/2</td><td>18.59</td><td>28.03</td><td>23.40</td><td>4.06%</td><td>1</td><td>18.61</td><td>**28.38**</td><td>23.25</td><td>0.42%</td></tr>
<tr><td>1/8</td><td>**18.92**</td><td>28.10</td><td>**24.11**</td><td>2.16%</td><td>0.5</td><td>**18.92**</td><td>28.10</td><td>**24.11**</td><td>0.33%</td></tr>
<tr><td>1/32</td><td>18.83</td><td>**28.39**</td><td>24.25</td><td>**0.23**%</td><td>0.25</td><td>18.56</td><td>27.65</td><td>24.09</td><td>**0.26**%</td></tr>
<tr><td>*CLIP*</td><td>*14.03*</td><td>*19.86*</td><td>*21.48*</td><td>*844.48 ms*</td><td>*CLIP*</td><td>*14.03*</td><td>*19.86*</td><td>*21.48*</td><td>*75.91 GiB*</td></tr>
</table>

form them around 60K encoder training iterations for both scales. We hypothesize that after this point, the encoder's output features stabilize sufficiently, thereby enhancing amortization optimization and enabling AMORLIP to exhibit superior performance gains. Overall, AMORLIP demonstrates faster convergence and increased relative efficiency at larger scales.

**Lightweight amortization**    Table 2 further quantifies the additional time and memory overhead of AMORLIP relative to CLIP by examining critical overhead-related factors. Specifically, when the amortization network is updated less frequently (lower $T_{\text{online}}^{-1}$) and utilizes fewer parameters (smaller dimension factor $f_d$), the extra GPU time and memory overhead become effectively minimized and eventually negligible (with only 0.26% higher memory usage and 0.23% additional training time), all while consistently outperforming the baseline in downstream tasks. Interestingly, reduced amortization network complexity or update frequency generally corresponds to improved model performance compared to larger amortization networks. A potential explanation is that smaller networks may mitigate overfitting during the amortization stage. Furthermore, this highlights that AMORLIP provides an effective representation for partition functions, thus achieving robust performance even with lightweight amortization implementations.

## 4.3 Ablation Studies

In the ablation studies, we evaluate the impact of several key factors associated with amortization. Unless otherwise specified, we adopt the medium-scale setting with the l2-log objective as in Table 1.

**Impact of stability techniques**    Figure 4 evaluates the impact of the two stability techniques introduced in Section 3.3. As shown in Figure 4a, using a larger EMA factor for the target model generally enhances both classification and overall performance. A larger EMA factor smooths updates of the target model parameters, which effectively preserves valuable information from previous batches. Figure 4b further examines the effect of different combination weights ($\beta_T$) for amortization based on previous-epoch information. A moderately high value of $\beta_T$, such as $\beta_T = 0.8$, facilitates amortization learning and particularly improves retrieval performance. However, excessively large values (e.g., $\beta_T = 1.0$) hinder effective updates from new batches and negatively affect overall performance. The ablations presented in Figure 4b suggest that $\beta_T = 0.8$ provides the best balance with optimal overall performance.

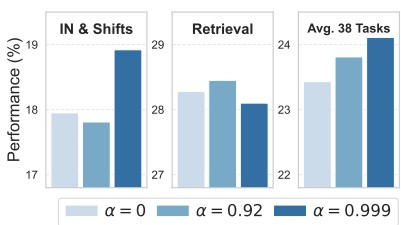

(a) Target EMA factor $\alpha$ in Eq. (11)

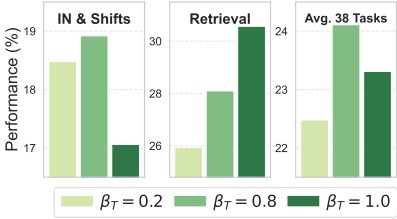

(b) Combination weight $\beta_T$ in Eq. (12)

Figure 4: Performance (%) of AMORLIP with different values of parameter $\alpha$ and $\beta_T$ in stability techniques.

**Amortization objective**    We compare two proposed amortization objectives in Section 3.2. As shown in Table 1 and Figure 3, the l2-log objective exhibits slightly better performance and smoother training dynamics than the $f$-divergence objective at both scales. As discussed in Sec-

tion 3.2, although the l2-log objective introduces bias, it effectively reduces variance. This indicates that the proposed parameterization of AMORLIP can successfully amortize the target partition function even under a biased objective. Additionally, the reduction in variance contributes to improved training performance.

**Amortization target** By default, the amortization objective targets the entire partition function, incorporating both positive and negative samples from each minibatch. Table 3 ablates the effect of amortizing only the partition functions computed from negative samples while using positive samples directly from the current batch. Results demonstrate that amortizing the full partition function consistently leads to improved performance across all three task groups. This outcome likely arises because positive samples typically yield higher similarity scores and may significantly influence the magnitude of the partition function.

Table 3: Performance (%) with amortizing only negative samples (Neg.) vs. the entire batch (Pos.+Neg.).

| Amor. Target | IN&Shifts | Ret. | Avg. 38 |
|---|---|---|---|
| Neg. | 18.02 | 27.92 | 23.17 |
| Pos. + Neg. | **18.92** | **28.10** | **24.11** |

### 4.4 Empirical Guidelines for Hyperparameter Setup

Based on our empirical evaluation above, we summarize concise and practical guidelines for instantiating hyperparameters:

**Amortization loss:** We proposed two amortization loss variants: $f$-divergence (unbiased) and l2-log (biased but with reduced variance). We recommend the l2-log objective when the learnable temperature varies significantly during training (as in standard CLIP pretraining, temperature from 14.27 to 100 [51]). In scenarios with stable temperature (*e.g.*, fine-tuning), the unbiased $f$-divergence objective may perform better.

**EMA decay factor $\alpha$:** The EMA decay factor $\alpha$ determines how much past model information is retained. Theoretically, a larger $\alpha$ stabilizes model training by smoothing gradient estimation noise and effectively preserving information from previous batches. Based on the empirical results in Figure 4a, we recommend using a relatively large $\alpha$, typically 0.999, which achieves the best results in small or medium scales.

**Combination weight $\beta_T$:** Empirically, a moderately high value of $\beta_T$, such as $\beta_T = 0.8$, facilitates amortization learning and particularly improves retrieval performance. However, excessively large values (*e.g.*, $\beta_T = 1.0$) may hinder effective updates from new batches and negatively affect overall performance. Figure 4b suggests that $\beta_T = 0.8$ provides the best balance with optimal overall performance.

**Update frequency $T_{\text{online}}^{-1}$:** The amortization network should be updated less frequently than the encoder models (Table 2a). Empirically, a frequency of $T_{\text{online}}^{-1} \leqslant \frac{1}{8}$ offers an optimal balance between performance and computational overhead for small and medium scales.

**Capacity ($f_d$) of the amortization network:** Reduced network complexity (smaller $f_d$) mitigates potential overfitting and minimizes memory overhead. Based on empirical results, we recommend setting $f_d$ to 0.5 or even 0.25 to achieve the best balance between efficiency and performance.

Finally, we emphasize that AMORLIP consistently outperforms baseline methods even with suboptimal hyperparameter settings. The effectiveness of AMORLIP helps reduce the necessity for extensive hyperparameter tuning.

## 5 Related Work

**Efficient CLIP training** In response to the rapid growth of data and model scales in CLIP training [39, 6, 62], several studies aim at improving training efficiency. Methods such as LiT [72], FLIP [43], and CLIPA [42] aim to lower computational complexity through strategies like model freezing, token masking, or resolution rescaling. Other approaches have modified the contrastive loss,

including the decoupled softmax loss in DCL [68], pairwise sigmoid loss in SigLIP [71], or aggregating local losses computed on subsets of each minibatch [11, 36, 12, 16]. However, these techniques often compromise downstream task performance or extend training durations. Another group of work leverages non-parametric estimation of the partition function, such as DeCL [11], SogCLR [70], iSogCLR [50], and NuCLR [64]. Furthermore, system-level approaches, such as modifications to gradient implementations [59, 49, 25] and distributed parallel training frameworks [63, 36, 15, 56, 16], have attempted to scale CLIP models with larger batch size or across large clusters of GPUs or TPUs. Nevertheless, these two types of approaches generally suffer from substantial overhead either in space (*e.g.*, maintaining large offline buffers) or time (*e.g.*, inter-device communication), potentially limiting their practical scalability and efficiency.

**Amortization in self-supervised learning**    In general self-supervised learning (SSL), many prominent methods reuse or approximate computationally expensive operations, such as computing large-scale similarity matrices or offload these tasks to auxiliary models. We refer to this overarching strategy as *amortization*. One common approach involves maintaining a memory buffer to store representations of previously encountered samples, thereby reducing per-batch computation and alleviating large batch size requirements [12]. Such memory banks are widely employed in contrastive and clustering-based SSL frameworks [67, 7, 3]. For instance, the MoCo family [30, 13], along with other related methods [61, 8], utilizes a queue-based buffer to amortize negative representation computations effectively. More recently, several methods [70, 50, 64] have adopted larger memory buffers that span the entire training set, amortizing the expensive partition function computation at a per-sample granularity. Another amortization strategy involves a momentum encoder updated slowly via an EMA model of the online encoder parameters [30, 13, 27]. The EMA model is also central to negative-sample-free methods such as BYOL [27]. Additionally, recent work [37] has proposed amortizing reconstruction tasks via meta-learning to further enhance SSL performance across multiple modalities. Unlike most existing methods with nonparametric amortization, the proposed AMORLIP directly learns the amortization targets by optimizing lightweight neural networks. This parametric amortization ensures greater flexibility while free of maintaining a large memory buffer.

**EBM Learning.**    Energy-based models (EBMs)[41] flexibly represent probability distributions using an energy function defined over data points. Specifically, a conditional EBM takes the form:

$$\mathbb{P}(y|x) = \frac{\exp(-f(x,y))}{Z(x)},$$

where the energy function $f(x, y)$, typically parameterized by deep neural networks, assigns lower energy values to more probable data points; the partition function $Z(x)$ ensures $\mathbb{P}(y|x)$ to be valid probability distributions. Training EBMs generally involves several techniques[60], such as MLE via MCMC sampling [34, 21], score matching [35, 46], and NCE [47, 28, 29]. To further improve efficiency, amortization techniques have been introduced into EBM training: SteinGAN[45] amortizes negative sample generation for MLE with a jointly trained sampler; ADE [17] leverages a primal-dual perspective on MLE to learn an efficient sampling strategy for exponential family distributions; and ALOE [19] introduces an amortized sampler inspired by local search to estimate gradients for EBMs on discrete structured data efficiently. Recently, CLIP-JEM [24] introduces an image-text joint-energy function in the CLIP representation space to enable text-to-image generation capabilities.

## 6    Conclusion

In this paper, we proposed AMORLIP, a novel amortization framework that effectively decouples the estimation of partition functions from minibatch-level optimization through lightweight neural networks. Extensive experimental results demonstrated that AMORLIP consistently outperforms existing CLIP-like baselines across 38 diverse downstream tasks, achieving substantial relative improvements of up to 12.24%. AMORLIP significantly enhances training efficiency and leads to more resource-efficient contrastive language-image pretraining.

## Acknowledgments and Disclosure of Funding

This work was supported in part by the ONR grant N000142512173, NSF grants ECCS: 2401391 and IIS: 2403240, Dolby support, and computing resources received from the National Supercomputing Center (CSCS) and the Swiss AI initiative.

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

# A  Limitations and Broader Impacts

## A.1  Limitations and Future Work

In this work, we proposed AMORLIP, a novel amortization paradigm to enhance CLIP training efficiency. Despite demonstrating effectiveness and efficiency, our proposed method exhibits several limitations:

**Resource constraints**   Due to computational resource limitations, our evaluations of AMORLIP and baseline methods were constrained to datasets with up to ten million samples and models up to the scale of ViT-B/32. Although experimental results at these scales consistently demonstrate the advantages of our method, further evaluation at billion-scale datasets and larger backbone models could better highlight AMORLIP's scalability and efficiency. We plan to address this in our future work.

**Data privacy and licensing**   We pretrained and evaluated AMORLIP using publicly available datasets in compliance with their respective licenses and intended use policies. Nevertheless, given the extensive scale of these datasets comprising millions of text-image pairs collected from the web, there remains a potential risk of encountering unintended or unfiltered content. This could inadvertently lead to privacy concerns or inadvertent exposure of sensitive information.

## A.2  Broader Impacts

**Potential positive societal impacts**   The proposed AMORLIP addresses an important challenge in contrastive language-image pretraining, *i.e.*, the extensive computational resource requirements that have limited accessibility and scalability. As demonstrated both theoretically and empirically, AMORLIP significantly reduces reliance on large batch sizes through efficient amortization techniques. Consequently, AMORLIP facilitates effective and efficient pretraining of vision-language models and enables a wider range of individuals and organizations to develop and deploy high-quality multimodal models across diverse downstream tasks, including cross-modal retrieval, zero-shot classification, and text-to-image synthesis.

**Potential negative societal impacts**   However, the proposed method may also lead to misleading results in downstream applications. Despite achieving superior representation performance compared to baselines, AMORLIP does not guarantee perfect accuracy or recall in downstream tasks. For example, when deployed in text-to-image retrieval scenarios, AMORLIP could potentially return mismatched or inappropriate images, leading to unintended consequences such as privacy leakage or exposure to harmful content. We therefore recommend deploying AMORLIP alongside robust data privacy protection and content moderation tools to effectively mitigate these risks.

## A.3  Ethical statements

When conducting research presented in the paper, we have fully conformed with the NeurIPS Code of Ethics. Additionally, our use of all models and datasets strictly adheres to their corresponding licenses and usage guidelines.

## B Theoretical Derivations

### B.1 Derivation of Random Features in Eq. (7)

We begin by rewriting (1) with the following equivalent energy-based parameterization [54, 73]: Since each $\psi_l(u_l)$ is $\ell_2$-normalized, we have $\|\psi_l(u_l)\|^2 = 1$. Then, one can write:

$$\mathbb{P}(u_l|u_{l'}) \propto \mathbb{P}(u_l) \exp\left(\tau \psi_l(u_l)^\top \psi_{l'}(u_{l'})\right)$$

$$= \mathbb{P}(u_l) \exp\left(\frac{\tau \|\psi_l(u_l)\|^2}{2} + \frac{\tau \|\psi_{l'}(u_{l'})\|^2}{2}\right) \exp\left(-\frac{\|\sqrt{\tau}\psi_l(u_l) - \sqrt{\tau}\psi_{l'}(u_{l'})\|^2}{2}\right)$$

$$= \mathbb{P}(u_l) \exp\left(-\frac{\|\sqrt{\tau}\psi_l(u_l) - \sqrt{\tau}\psi_{l'}(u_{l'})\|^2}{2}\right) \exp(\tau),$$

where $\exp\left(-\|\sqrt{\tau}\psi_l - \sqrt{\tau}\psi_{l'}\|^2/2\right)$ is the Gaussian kernel and induces the spectral decomposition by applying random Fourier features. Denote $\delta_{l,l'} = (\sqrt{\tau}\psi_l - \sqrt{\tau}\psi_{l'}) \in \mathbb{R}^d$ for notation simplicity. One can write:

$$\exp\left(-\frac{\|\sqrt{\tau}\psi_l - \sqrt{\tau}\psi_{l'}\|^2}{2}\right)$$

$$= \exp\left(-\frac{(\delta_{l,l'})^\top \delta_{l,l'}}{2}\right)$$

$$= (2\pi)^{-\frac{d}{2}} \exp\left(-\frac{\|\delta_{l,l'}\|^2}{2}\right) \int_{\mathbb{R}^d} \exp\left(-\frac{\|\omega - \mathbf{i}\delta_{l,l'}\|^2}{2}\right) d\omega$$

$$= (2\pi)^{-\frac{d}{2}} \int_{\mathbb{R}^d} \exp\left(-\frac{\|\omega\|^2}{2} + \mathbf{i}\omega^\top \delta_{l,l'}\right) d\omega$$

$$= \mathbb{E}_{\omega \sim \mathcal{N}(0, \mathbf{I}_d)}\left[\exp\left(\mathbf{i}\omega^\top \delta_{l,l'}\right)\right]$$

$$= \mathbb{E}_{\omega \sim \mathcal{N}(0, \mathbf{I}_d)}\left[\exp\left(\mathbf{i}\sqrt{\tau}\omega^\top \psi_l\right) \exp\left(-\mathbf{i}\sqrt{\tau}\omega^\top \psi_{l'}\right)\right],$$

which leads to Eq. (7).

### B.2 Derivation of Divergence Objective in Eq. (8)

With the definition of $D_f(\cdot, \cdot)$ in Section 2, one can write:

$$\ell_{\text{amor, }f\text{-div}}$$

$$= \mathbb{E}_{\mathbb{P}(u_l)}\left[D_f\left(\mathbb{Q}(u_{l'}|u_l), \mathbb{P}(u_{l'}|u_l)\right)\right]$$

$$= \mathbb{E}_{\mathbb{P}(u_l)}\left[\int \mathbb{P}(u_{l'}) \frac{\exp\left(\tau \psi_l(u_l)^\top \psi_{l'}(u_{l'})\right)}{Z_l(u_l)} f\left(\frac{\exp\left(\tau \psi_l(u_l)^\top \psi_{l'}(u_{l'})\right)/\lambda_{\theta_l}(u_l)}{\exp\left(\tau \psi_l(u_l)^\top \psi_{l'}(u_{l'})\right)/Z_l(u_l)}\right) d\mu(u_{l'})\right]$$

$$= \mathbb{E}_{\mathbb{P}(u_l)\mathbb{P}(u_{l'})}\left[\frac{\exp\left(\tau \psi_l(u_l)^\top \psi_{l'}(u_{l'})\right)}{Z_l(u_l)} f\left(\frac{Z_l(u_l)}{\lambda_{\theta_l}(u_l)}\right)\right].$$

where $\mu$ is a base measure for $u_{l'}$.

## B.3 Connection between $\ell_{\text{amor}, f\text{-div}}$ and $\ell_{\text{amor}, \text{l2-log}}$

We show that Eq. (10) can be included into the family of $f$-divergence in Eq. (8) with $f(t) = \frac{1}{2}(\log t)^2$:

$$\mathbb{E}_{\mathbb{P}(u_l)\mathbb{P}(u_{l'})} \left[ \exp\left( \tau \psi_l\left(u_l\right)^\top \psi_{l'}\left(u_{l'}\right) - \log Z_l\left(u_l\right) \right) f\left( \frac{Z_l\left(u_l\right)}{\lambda_{\theta_l}\left(u_l\right)} \right) \right]$$

$$= \frac{1}{2}\mathbb{E}_{\mathbb{P}(u_l)} \left[ \frac{\mathbb{E}_{\mathbb{P}(u_{l'})}\left[ \exp\left( \tau \psi_l\left(u_l\right)^\top \psi_{l'}\left(u_{l'}\right) \right) \right]}{Z_l\left(u_l\right)} \left\| \log\left( \frac{Z_l\left(u_l\right)}{\lambda_{\theta_l}\left(u_l\right)} \right) \right\|^2 \right]$$

$$= \frac{1}{2}\mathbb{E}_{\mathbb{P}(u_l)} \left[ \left\| \log \lambda_{\theta_l}\left(u_l\right) - \log Z_l\left(u_l\right) \right\|^2 \right],$$

which recovers Eq. (10).

## B.4 Connection between $\ell_{\text{amor}, kl\text{-div}}$ and $\ell_{\text{amor}, \text{l2-log}}$

We demonstrate that $\ell_{\text{amor}, \text{l2-log}}$ closely approximates the KL divergence up to second order within the general $f$-divergence framework. For simplicity of notation, we denote $\mathbb{P}_0(x)$ as the fixed probability distribution and $\mathbb{Q}_\theta(x)$ as the distribution parameterized by $\theta$. We assume $\mathbb{Q}_{\theta_0}$ closely approximates $\mathbb{P}_0$ at $\theta = \theta_0$, i.e., $\mathbb{P}_0 = \mathbb{Q}_{\theta_0}$. Additionally, we define the score function $s_\theta(x)$ and the Fisher information matrix $G_\theta$ for $\mathbb{Q}_{\theta_0}$ as:

$$s_\theta(x) = \nabla_\theta \log q_\theta(x), \quad G_\theta = \mathbb{E}_{q_\theta}\left[s_\theta(x)s_\theta(x)^\top\right].$$

Considering $D_f(\mathbb{Q}_\theta, \mathbb{P}_0)$ as a scalar function of $\theta$ that is at least twice continuously differentiable in a neighborhood around the point $\theta_0$, we apply the second-order Taylor expansion to obtain:

$$D_f(\theta) = D_f(\theta_0) + (\theta - \theta_0)^\top \nabla_\theta D_f(\theta_0) + \frac{1}{2}(\theta - \theta_0)^\top \left[\nabla_\theta^2 D_f(\theta_0)\right](\theta - \theta_0) + \mathcal{O}(\|\theta\|^3). \quad (13)$$

Here, the first term vanishes since $D_f(\theta_0) = f(\frac{\mathbb{Q}_{\theta_0}}{\mathbb{P}_0}) = f(1) = 0$. Then, we show that the first-order gradient in Eq. (13) is also zero at $\theta_0$. Specifically, one can write:

$$\nabla_\theta D_f(\mathbb{Q}_\theta, \mathbb{P}_0) = \nabla_\theta \int p_0(x) f\left( \frac{q_\theta(x)}{p_0(x)} \right) d\mu(x)$$

$$= \int p_0(x) \nabla_\theta \left[ f\left( \frac{q_\theta(x)}{p_0(x)} \right) \right] d\mu(x)$$

$$= \int p_0(x) f'\left( \frac{q_\theta(x)}{p_0(x)} \right) \frac{\nabla_\theta q_\theta(x)}{p_0(x)} d\mu(x) \quad (14)$$

$$= \int q_\theta(x) f'\left( \frac{q_\theta(x)}{p_0(x)} \right) s_\theta(x) d\mu(x)$$

$$= \mathbb{E}_{q_\theta}\left[ f'\left( \frac{q_\theta(x)}{p_0(x)} \right) s_\theta(x) \right].$$

Evaluating Eq.(14) at $\theta_0$ yields:

$$\nabla_\theta D_f(\theta_0) = f'(1) \cdot \mathbb{E}_{q_\theta}\left[s_\theta(x)\right]$$

$$= f'(1) \cdot \int q_\theta(x) \frac{\nabla_\theta q_\theta(x)}{q_\theta(x)} d\mu(x)$$

$$= f'(1) \cdot \nabla_\theta \left( \int q_\theta(x) d\mu(x) \right) = 0.$$

Similarly, we derive the Hessian matrix in Eq. (13):

$$\nabla_\theta^2 D_f(\mathbb{Q}_\theta, \mathbb{P}_0) = \int \left[ f''\left( \frac{q_\theta(x)}{p_0(x)} \right) \frac{1}{p_0(x)} \left( q_\theta(x) s_\theta(x) \right) \left( q_\theta(x) s_\theta(x) \right)^\top \right.$$

$$\left. + f'\left( \frac{q_\theta(x)}{p_0(x)} \right) q_\theta(x) \left( s_\theta(x) s_\theta(x)^\top + \nabla_\theta s_\theta(x) \right) \right] d\mu(x) \quad (15)$$

$$= \int q_\theta(x) \left[ f''\left( \frac{q_\theta}{p_0} \right) \frac{q_\theta}{p_0} s_\theta s_\theta^\top + f'\left( \frac{q_\theta}{p_0} \right) \left( s_\theta s_\theta^\top + \nabla_\theta s_\theta \right) \right] d\mu(x).$$

Evaluating Eq. (15) at $\theta_0$ yields:

$$\nabla_\theta^2 D_f(\theta_0) = f''(1)\mathbb{E}_{q_{\theta_0}}\left[s_{\theta_0}s_{\theta_0}^\top\right] + f'(1)\left(\mathbb{E}_{q_{\theta_0}}\left[s_{\theta_0}s_{\theta_0}^\top\right] + \mathbb{E}_{q_{\theta_0}}\left[\nabla_\theta s_\theta\big|_{\theta_0}\right]\right)$$
$$= f''(1)G_{\theta_0} + f'(1)\left(G_{\theta_0} - G_{\theta_0}\right) = f''(1)G_{\theta_0},$$

where $\mathbb{E}_{q_{\theta_0}}\left[\nabla_\theta s_\theta\big|_{\theta_0}\right] = -G_{\theta_0}$ due to Bartlett's second identity [5]. Therefore, for any variant within the $f$-divergence family, Eq. (13) simplifies to:

$$D_f(\theta) = \frac{1}{2}f''(1)(\theta - \theta_0)^\top G_{\theta_0}(\theta - \theta_0) + \mathcal{O}(\|\theta\|^3). \tag{16}$$

For $f_{\text{kl-div}}(t) = t\log t$ and $f_{\text{l2-log}}(t) = \frac{1}{2}(\log t)^2$, it is straightforward to verify that $f''_{\text{kl-div}}(1) = f''_{\text{l2-log}}(1) = 1$. Thus, the l2-log estimator closely approximates the KL divergence up to second order.

## C  Detailed Experimental Setup

For training the encoders, we employ the AdamW optimizer [40] with a learning rate of $1 \times 10^{-3}$ for the medium-scale setting and $4 \times 10^{-4}$ for the large-scale setting. For updating the amortization network, we use the Adam optimizer [40] universally set at a learning rate of $1 \times 10^{-3}$. Following the temperature scaling technique proposed in [66], we rescale the contrastive loss and adopt an additional regularizer $\rho$, i.e., $\ell_{\text{NCE, rescaled}} = \ell_{\text{NCE}}/\texttt{stop\_grad}(\tau) + \rho/\tau$. Consistent with [66], we set $\rho = 6.5$ for the medium-scale setting and $\rho = 8.5$ for the large-scale setting. Additionally, we introduce temperature annealing to further improve learning efficiency. Specifically, we reset $\rho$ to $-8.0$ during the last quarter of epochs in the medium-scale setting and to $-8.5$ during the last third of epochs in the large-scale setting. For AMORLIP$_{f\text{-div}}$, we add l2-log loss as regularizer with coefficient of 0.1 and sweep between two divergence formulations $\{kl, js\}$ and report the best results in Table 1.

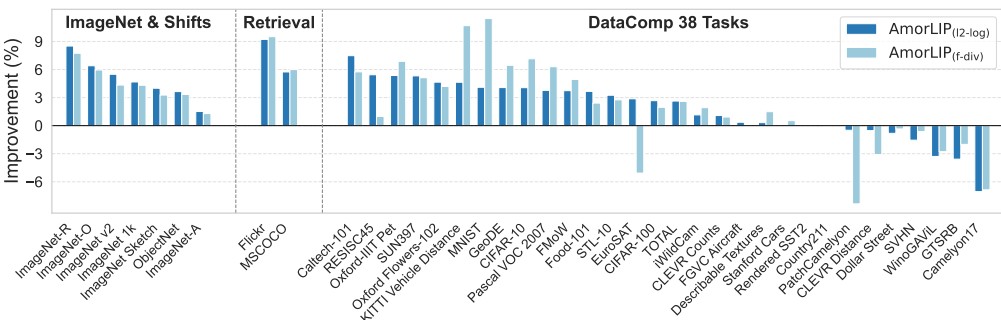

Figure 5: Breakdown of absolute improvement (%) made by AMORLIP over CLIP model on all 38 DataComp Tasks [23] under medium scale setting.

## D  More DataComp Results Breakdown

Figure 5 showcases the absolute improvement delivered by AMORLIP over CLIP across all Datacomp tasks in the medium setting.

