# OpenReview forum: "AmorLIP: Efficient Language-Image Pretraining via Amortization"
_NeurIPS.cc/2025/Conference — NeurIPS 2025 poster_

### Official Review · Reviewer_ePbH · 2025-06-25

**Clarity:** 4
**Significance:** 3
**Originality:** 3
**Rating:** 4
**Confidence:** 4

**Summary:**

AMORLIP proposes a contrastive language-image pretraining framework that amortizes the partition function using a lightweight network, in order to eliminate the rigid dependence on large batches of negatives as in CLIP training. From the perspective of energy-based models, the partition function Z is expressed as a linear form of random features, and a small MLP is used to amortize its computation in a two-stage alternating optimization scheme. The framework also introduces two objective variants—f-div and l2-log—to balance the bias-variance trade-off. Experiments are conducted on both CC-3M + ResNet-50 and CC-12M + ViT-B/32 settings, covering 38 downstream DataComp tasks. Compared to CLIP, AMORLIP achieves up to a 12.24% improvement in average zero-shot performance.

**Questions:**

In visual self-supervised contrastive pretraining, it has traditionally been assumed that a large number of negative samples is necessary. However, some studies [1,2] have shown that if certain hyperparameters—such as the temperature coefficient, loss scaling, or gradient weighting—are properly tuned, contrastive learning can still perform as well as or even better than the original large-batch setting, even with only a few negatives per pair.
Do you think similar effects could be achieved in vision-language contrastive pretraining? If not, what differences do you think might account for this?

[1] EqCo: Equivalent Rules for Self-supervised Contrastive Learning

[2] Are All Negatives Created Equal in Contrastive Instance Discrimination?

**Ethical Concerns:**

["NO or VERY MINOR ethics concerns only"]

**Final Justification:**

I have carefully read the authors’ response and the other reviewers’ comments, and my questions have been largely resolved. Overall, I find this paper solid compared with others in its field, so I will keep my score unchanged.

**Limitations:**

yes

**Paper Formatting Concerns:**

- The l2-log objective is inconsistently referred to as ℓ₂-log, l2-log, and l2-log loss across the paper.
- The same reference for BYOL (Grill et al.) appears repeatedly as both [26] and [27].

**Quality:**

3

**Strengths And Weaknesses:**

### Strengths
- The proposed method is both novel and well-motivated. By parameterizing the partition function in contrastive learning using spectral decomposition and a lightweight MLP, it avoids reliance on large batches and all-gather operations. Although the NCE community has long treated Z as a parameter and approximated it with conditional networks, AMORLIP is the first to effectively transfer this idea to large-scale cross-modal contrastive learning, resolving distributed bottlenecks via "two-stage amortization + sparse all_gather."
- The experimental evaluation is thorough, spanning 38 downstream tasks across two scales, and comparing systematically with four efficient CLIP baselines (e.g., SigLIP, SogCLR, FastCLIP).

### Weaknesses
- The use of two amortization losses and multiple hyperparameters (α, β_T, T_online, f_d) increases the tuning burden, which may hinder scalability in large-scale pretraining. Some ablation studies show the performance is sensitive to β_T. It would be helpful to provide empirical guidelines for setting these hyperparameters based on data scale and model size.
- Only zero-shot classification and retrieval results are reported, with no fine-tuning experiments or analysis.

---

> ### Author Rebuttal · Authors · 2025-07-30
>
> We sincerely appreciate the Reviewer for providing valuable feedback. Please find the responses below:
>
> ## [Empirical guidelines for hyperparameter setup]
>
> >  It would be helpful to provide empirical guidelines for setting these hyperparameters based on data scale and model size.
>
> Thank you for this valuable feedback. Based on our empirical evaluation (Sec. 4), we provide concise and practical guidelines for choosing these hyperparameters:
>
> -   **Amortization loss:** We proposed two amortization loss variants: f-divergence (unbiased) and l2-log (biased but with reduced variance). We recommend the l2-log objective when the learnable temperature varies significantly during training (as in standard CLIP pretraining, temperature from ~14.27 to 100 [1]). In scenarios with stable temperature (e.g., fine-tuning), the unbiased f-divergence objective may perform better.
>
>
> -   **EMA decay factor $\alpha$:** The EMA decay factor $\alpha$ determines how much past model information is retained. Theoretically, a larger $\alpha$ stabilizes model training by smoothing gradient estimation noise and effectively preserving information from previous batches. Based on the empirical results in Figure 4(a), we recommend using a relatively large $\alpha$ value, typically 0.999 achieves the best results in small or medium scales.
>
>
> -   **Combination weight $\beta_T$:**  Empirically, a moderately high value of $\beta_T$, such as $\beta_T=0.8$, facilitates amortization learning and particularly improves retrieval performance. However, excessively large values (e.g., $\beta_T=1.0$) may hinder effective updates from new batches and negatively affect overall performance. Figure 4b suggests that $\beta_T=0.8$ provides the best balance with optimal overall performance.
>
>
> -   **Update frequency $T_{\text{online}}^{-1}$ for amortization network:** The amortization network should be updated less frequently than the encoder models (Table 2a). Empirically, a frequency of $T_{\text{online}}^{-1}\leq\frac{1}{8}$ offers an optimal balance between performance and computational overhead for small and medium scales.
>
>
> -   **Capacity ($f_d$) of the amortization network:** Reduced network complexity (smaller $f_d$) mitigates potential overfitting and minimizes memory overhead. Based on empirical results, we recommend setting $f_d$ to 0.5 or even 0.25 to achieve the best balance between efficiency and performance.
>
> Finally, we emphasize that AmorLIP consistently outperforms baseline methods even with suboptimal hyperparameter settings. The effectiveness of AmorLIP helps reduce the necessity for extensive hyperparameter tuning.
>
>
>
> ## [Additional fine-tuning experiments]
>
> >  Only zero-shot classification and retrieval results are reported, with no fine-tuning experiments or analysis.
>
>
> Thank you for this suggestion. Our evaluations primarily focused on zero-shot classification and retrieval tasks using the standard DataComp benchmark, consistent with prior works [1-3], to clearly demonstrate AmorLIP’s effectiveness in foundational representation learning.
>
> As suggested by the Reviewer, incorporating fine-tuning experiments would indeed make our evaluation more comprehensive and provide valuable insights into how the learned representations transfer when fine-tuned on downstream tasks. However, our work, consistent with prior studies [1-3], primarily focuses on developing a high-performing pre-training approach that is dataset-agnostic. Fine-tuning stage adapts representations to each specific dataset, which could potentially compensate for or mask weaknesses in learning general and robust representations during pre-training. Additionally, the complexities introduced by fine-tuning tasks might obscure direct and clear comparisons of foundational representation quality. Thus, we intentionally chose to maintain a clear and direct evaluation scope using only zero-shot classification and retrieval tasks.
>
> Nevertheless, we acknowledge the importance of fine-tuning evaluations and will actively pursue them in future extensions of our work.
>
> ## [Transferability of Hyperparameter Tuning in AmorLIP]
>
> >   Do you think similar effects could be achieved in vision-language contrastive pretraining? If not, what differences do you think might account for this?
>
> We thank the Reviewer for this insightful question. Indeed, both vision-language (VL) and visual self-supervised learning (SSL) rely fundamentally on contrastive (NCE) loss; thus, insights such as loss scaling and temperature tuning are potentially valuable across both domains. However, we would like to highlight a crucial difference specific to VL pretraining, which makes directly transferring these techniques non-trivial.
>
> In VL datasets, one image can correspond to multiple valid captions, and multiple images can share similar descriptions, leading to a significant issue of false negatives [4]. These false negatives (semantically related pairs mistakenly treated as unrelated) introduce complexities beyond the visual SSL setting. Consequently, directly applying aggressive loss scaling or gradient weighting as in visual SSL [5,6] might unintentionally amplify the impact of false negatives, potentially degrading VL model performance.
>
> That said, certain principles from SSL remain beneficial for VL contrastive learning. Specifically, in AmorLIP, our amortized partition function can be viewed as an adaptive, gradient-free weighting mechanism. The amortization network dynamically learns these weights based on input samples and carefully designed amortization objectives, to inherently capture cross-modal relations. This suggests that adaptive loss scaling and gradient weighting can still be beneficial for VL, provided they account explicitly for cross-modal complexities.
>
> Furthermore, we emphasize that techniques such as loss scaling and temperature tuning are orthogonal to our method, and thus could potentially be combined with AmorLIP to further boost performance. For instance, as detailed in Appendix D, we have already incorporated a rescaled contrastive loss to enhance training stability: $\ell_{\text{NCE, rescaled}}=\ell_{\text{NCE}}/\text{stopgrad}(\tau)+\rho/\tau$, where $\tau$ is the learnable temperature and $\rho$ is the scaling factor, following best practices from [2].
>
> In summary, while our framework provides a more generalized methodology for VL pretraining, integrating insights from well-established visual SSL techniques remains promising. We plan to explore these connections further in our future work.
>
>
>
> ## [Paper formatting]
>
> Thank you for pointing out the typos. We will correct them in our updated version.
>
>
>
> ##  References
>
> [1] Radford, Alec, et al. "Learning transferable visual models from natural language supervision." International conference on machine learning. PmLR, 2021.
>
> [2] Wei, Xiyuan, et al. "Fastclip: A suite of optimization techniques to accelerate clip training with limited resources." arXiv preprint arXiv:2407.01445 (2024).
>
> [3] Zhai, Xiaohua, et al. "Sigmoid loss for language image pre-training." Proceedings of the IEEE/CVF international conference on computer vision. 2023.
>
> [4] Bulat, Adrian, Yassine Ouali, and Georgios Tzimiropoulos. "FFF: Fixing Flawed Foundations in contrastive pre-training results in very strong Vision-Language models." Proceedings of the IEEE/CVF Conference on Computer Vision and Pattern Recognition. 2024.
>
> [5] EqCo: Equivalent Rules for Self-supervised Contrastive Learning
>
> [6] Are All Negatives Created Equal in Contrastive Instance Discrimination?

---

> > ### Comment · Reviewer_ePbH · 2025-08-08
> >
> > I have carefully read the authors’ response and the other reviewers’ comments, and my questions have been largely resolved. Overall, I find this paper solid compared with others in its field, so I will keep my score unchanged.

---

### Official Review · Reviewer_g7oi · 2025-07-02

**Clarity:** 3
**Significance:** 2
**Originality:** 3
**Rating:** 4
**Confidence:** 3

**Summary:**

This papers focus on conventional image language pretraining stage. Different from contrastive learning based CLIP model with larger GPU memory and batch size, this work focus on improving the training efficiency and performance. The main idea is sourced from energy-based models.

**Questions:**

1. Can you provide more discussion about the difference between Sigmoid Loss for Language Image Pre-Training and this work in technique?
2. Also, provide more discussion with Provable Guarantees for Self-Supervised Deep Learning with Spectral Contrastive Loss.

**Ethical Concerns:**

["NO or VERY MINOR ethics concerns only"]

**Final Justification:**

The paper sets out to challenge prominent models like CLIP and SigLIP and the work is interesting. But the CC datasets are somewhat dated and the absence of code limits the ability to fully assess the generalizability and broader impact. I tend to accept this work.

**Limitations:**

Yes

**Paper Formatting Concerns:**

None.

**Quality:**

3

**Strengths And Weaknesses:**

Pros:

1. Strong Performance Gains:
The proposed method demonstrates significant improvements, particularly in image classification and retrieval tasks. It is extensively evaluated across all DataComp benchmarks, providing a thorough and convincing performance comparison.
2. Comprehensive Baseline Comparisons:
Beyond standard CLIP, the work includes fair and detailed comparisons against recent strong baselines such as FastCLIP, SigLIP, and SogCLR, highlighting the robustness and effectiveness of the approach.
3. Innovative Use of Energy-Based Models:
The integration of an energy-based function into the CLIP framework is a novel and promising direction, potentially offering new insights into contrastive learning and model efficiency.
Cons:

Scalability on Larger, High-Quality Datasets Not Demonstrated:
1. A key concern is the limited scale and quality of the pretraining datasets. While CC3M and CC12M are used with rule-based filtering, they still contain significant noise and lack the quality of more modern datasets like DataComp-1B, LAION-COCO, or ShareGPT4V. To convincingly demonstrate scalability and robustness, it is recommended to pretrain on at least 100M+ high-quality samples or preferably on the full DataComp-1B set. Addressing this limitation could significantly enhance the impact and credibility of the work.
2. Limited Downstream Task Diversity:
The evaluation is primarily focused on classification and retrieval tasks within the DataComp benchmark. To better position this method as a general-purpose vision encoder, it would be valuable to include results on broader multimodal tasks such as visual question answering (VQA), image captioning, or referring expression grounding.
3. Different model parameters scale is also not explored.

---

> ### Author Rebuttal · Authors · 2025-07-30
>
> Thank you for the insightful comments. Please find our responses below:
>
> ## [Experiments using larger pretraining datasets]
>
> >  To convincingly demonstrate scalability and robustness, it is recommended to pretrain on at least 100M+ high-quality samples or preferably on the full DataComp-1B set. Addressing this limitation could significantly enhance the impact and credibility of the work.
>
> In our submission, we comprehensively evaluated AmorLIP and baselines using medium- (CC3M) and large-scale (CC12M) datasets across 38 tasks. As shown in Table 1 and Figure 3, AmorLIP consistently outperforms all baselines and demonstrates robust and transferable gains, despite the noise present in the datasets. Specifically, AmorLIP surpasses CLIP by 2.63% (CC3M) and 3.01% (CC12M) on average across DataComp tasks.
>
> To further investigate scalability to larger datasets, as the reviewer suggested, we are working on validating the effectiveness of our proposed method with much larger-scale datasets. We have initiated additional experiments on LAION-400M with the x-large setup from [1] (listed in the following Table) and plan to continue on the full DataComp-1B dataset afterwards.
>
> | Visual Encoder | Training Epochs | Batch Size | GPUs |
> |----------------|------------------|------------|----------|
> | ViT-B/16 | 42 Epochs | 640×8 | 8 × H100 |
>
> Unfortunately, despite our best efforts to secure additional compute resources, our current computational constraints mean these experiments require more than four weeks and cannot be completed during the rebuttal period. We appreciate the Reviewer for this suggestion and commit to finalizing these experiments and including detailed results and analyses in an updated version of our paper if permitted.
>
> ## [Downstream task diversity]
>
> >   To better position this method as a general-purpose vision encoder, it would be valuable to include results on broader multimodal tasks such as visual question answering (VQA), image captioning, or referring expression grounding.
>
> Thank you for this valuable feedback. Our evaluations primarily focused on zero-shot classification and retrieval tasks using the standard DataComp benchmark, consistent with prior works [1-3], to clearly demonstrate AmorLIP’s effectiveness in foundational representation learning.
>
> We agree with the Reviewer that evaluating broader multimodal tasks, such as visual question answering (VQA), image captioning, or referring expression grounding, can further establish our method as a general-purpose vision-language encoder. However, since these tasks typically require additional generative modeling or specialized fine-tuning steps, they introduce complexities that might obscure direct comparisons in foundational representation pretraining. Thus, we chose to maintain a clear and direct evaluation scope within classification and retrieval tasks.
>
> Nevertheless, we acknowledge the importance of extending our evaluation to general multimodal tasks and will actively explore these in our future work.
>
> ## [Different model scales]
>
> >   Different model  parameters scale is also not explored.
>
> Thank you for highlighting this important aspect. Following common practices in prior work [1], our experiments evaluated AmorLIP across two dataset scales (CC3M and CC12M) and two encoder architectures (ViT and ResNet). As shown in Table 1 and Figure 3, our results indicate that the advantages of AmorLIP consistently hold or even strengthen as model and dataset scales grow. For fairness, we kept all comparisons to baselines with the same model size. Additionally, we studied the impact of varying the parametric amortization model’s scale (Table 2b) and confirmed its robustness.
>
> We appreciate the Reviewer’s suggestion and acknowledge that exploring different encoder model parameter scales would further demonstrate the scalability and effectiveness of AmorLIP. Currently, we are actively scaling our experiments to larger datasets (e.g., LAION-400M), as mentioned previously, and we plan to systematically investigate varying encoder scales in future experiments.
>
> ## [Differences between AmorLIP and SigLIP[5]]
>
> >   Can you provide more discussion about the difference between Sigmoid Loss for Language Image Pre-Training and this work in technique?
>
> We thank the Reviewer for highlighting this important distinction. Below, we clarify and summarize the core differences more concisely:
>
> **Core difference (contrastive learning objective):** The fundamental difference between AmorLIP and SigLIP lies in their respective contrastive objectives for representation learning. Specifically, AmorLIP learns representations by parameterizing the target conditional distribution $\mathbb{P}(u_l\mid u_{l’})$ using a spectral factorization view from energy-based models (EBMs). This representation can be efficiently optimized through ranking-based noise contrastive estimation (ranking-based NCE), as adopted in CLIP. In contrast, SigLIP employs a binary classification-based NCE, implicitly assuming the partition function is independent of conditional inputs.
>
> **Learning complications:** As analyzed in [4], SigLIP’s binary NCE (classification-based) formulation requires an additional bias term for self-normalization and model consistency. This bias term can significantly affect performance, as shown in the original SigLIP paper ([5], Sec. 4.9). The observations also align with our experiments that AmorLIP shows consistent performance gain over SigLIP. In our experiments, we observed that after approximately 46k training steps, SigLIP’s contrastive validation loss began to increase, indicating potential overfitting or instability under small to medium-scale settings. We hypothesize this is due to the learning complication of the bias term under small/medium scale, aligning with empirical findings from [4]. In contrast, AmorLIP’s ranking-based NCE is inherently self-normalized and does not rely on an additional bias term, thus avoiding the stability and performance sensitivity observed in SigLIP.
>
> **Computational efficiency differences:** Conventional CLIP implementations rely heavily on the costly all_gather(·) operation at each step to estimate the partition function (Eq. (3)). SigLIP reduces this computational burden by dividing sample pairs into independent blocks, though at the expense of assuming an input-independent partition function. AmorLIP similarly enables efficient blockwise computation of the loss by parameterizing the partition function as a lightweight network dependent solely on the conditional inputs. This approach retains computational efficiency while preserving the theoretical advantages of ranking-based NCE.
>
> We will further highlight these key differences clearly in future paper updates.
>
> ## [Discussion with Spectral Contrastive Loss[6]]
>
> >   Also, provide more discussion with Provable Guarantees for Self-Supervised Deep Learning with Spectral Contrastive Loss.
>
> Thank you for raising this insightful comparison. Below, we concisely summarize the theoretical and empirical distinctions between AmorLIP and the spectral contrastive loss (SCL) from [6]:
>
> **Core difference (contrastive objective):** SCL derives its contrastive objective via an SVD-based factorization of the target conditional distribution $\mathbb{P}(u_l\mid u_{l’})$. The key difference between SCL and AmorLIP lies in the second term of their respective objectives. Specifically, AmorLIP employs an exponential operation (Eq. 5 in our paper), which effectively maps features into an infinite-dimensional space, enhancing model expressiveness. In contrast, SCL uses a quadratic term (Eq. 6 in [6]), expanding features into only a finite polynomial-dimensional space, which may limit expressiveness. Thus, SCL often requires significantly larger feature dimensions to achieve comparable performance (e.g., the feature dimension of 8192 is used in [6] for ImageNet classification.)
>
> **Gradient variance analysis:** The gradient of SCL’s quadratic term introduces an unnormalized weighting term $2\psi_l(u_l)^\top \psi_{l’}(u_{l’})$. This potentially causes high gradient variance and instability during optimization. AmorLIP, on the other hand, employs a softmax weighting via its NCE loss, resulting in bounded and more smoothed gradient estimates.
>
> **Empirical validation:** To further validate our analysis, we conducted ablation studies by replacing CLIP’s loss with SCL and pretrained on CC3M. The ImageNet-1K classification results below highlight SCL’s limited expressiveness and AmorLIP’s representation capability.
>
> | Model | SCL | CLIP | AmorLIP (l2-log) |
> |------------------|---------|---------|------------------|
> | Acc. | 11.64% | 16.84% | **21.50%** |
>
>
>
>
>
> ## Reference
>
> [1] Wei, Xiyuan, et al. "Fastclip: A suite of optimization techniques to accelerate clip training with limited resources." arXiv preprint arXiv:2407.01445 (2024).
>
> [2] Radford, Alec, et al. "Learning transferable visual models from natural language supervision." International conference on machine learning. PmLR, 2021.
>
> [3] Zhai, Xiaohua, et al. "Sigmoid loss for language image pre-training." Proceedings of the IEEE/CVF international conference on computer vision. 2023.
>
> [4] Ma, Zhuang, and Michael Collins. "Noise contrastive estimation and negative sampling for conditional models: Consistency and statistical efficiency." arXiv preprint arXiv:1809.01812 (2018).
>
> [5] Zhai, Xiaohua, et al. "Sigmoid loss for language image pre-training." Proceedings of the IEEE/CVF international conference on computer vision. 2023.
>
> [6] HaoChen, Jeff Z., et al. "Provable guarantees for self-supervised deep learning with spectral contrastive loss." Advances in neural information processing systems 34 (2021): 5000-5011.

---

> > ### Comment · Reviewer_g7oi · 2025-08-04
> >
> > The comparison with related work has been reasonably clarified.
> >
> >  I acknowledge the authors' constraints in conducting scaling-up experiments and exploring more downstream tasks. The paper sets out to challenge prominent models like CLIP and SigLIP, though these baselines are not evaluated directly on CC3M. But the CC datasets are somewhat dated and the absence of code limits the ability to fully assess the generalizability and broader impact. I will keep my current vote.

---

> > > ### Author Response · Authors · 2025-08-04
> > >
> > > Thank you for your insightful feedback. We’re glad our responses addressed your previous concerns. We are actively working on the larger-scale experiments and will include these results along with open-sourced code implementation in the updated version of our paper if permitted.

---

### Official Review · Reviewer_31TJ · 2025-07-02

**Clarity:** 3
**Significance:** 4
**Originality:** 3
**Rating:** 5
**Confidence:** 4

**Summary:**

This paper addresses a major bottleneck in Contrastive Language-Image Pretraining (CLIP): the immense computational cost and the requirement for extremely large batch sizes. To overcome this, the authors propose AMORLIP, an efficient pretraining framework that significantly reduces the dependency on large batches. The core idea is to "amortize" the most computationally expensive part of the contrastive loss—the partition function (the denominator in the softmax)—using lightweight neural networks. Extensive experiments on the 38 tasks of the DataComp benchmark show that AMORLIP not only matches but consistently and significantly outperforms standard CLIP and other efficiency-focused baselines.

**Questions:**

1. The paper compares two amortization objectives: an f-divergence-based objective and an L2-log objective. The L2-log objective appears to perform slightly better and more consistently. Is there a theoretical or intuitive reason to prefer one over the other? For instance, are there specific data regimes or training dynamics where the unbiased but potentially higher-variance f-divergence estimator might be advantageous?

2. The stability of the training relies on a target network for the amortization function, updated via EMA. How sensitive is the training process to the EMA decay factor α? Figure 4a shows that α=0.999 is best, but is there a "cliff" where a slightly different value leads to instability, or is the performance relatively robust within a certain range?

3. The core idea is to learn a parametric model (λ_θ) of the partition function. How does the capacity of this model (i.e., the MLP size) affect performance? The ablation in Table 2b suggests that smaller networks can be better, which is somewhat counter-intuitive. Could you elaborate on the hypothesis that this mitigates overfitting during the amortization stage?

**Ethical Concerns:**

["NO or VERY MINOR ethics concerns only"]

**Limitations:**

yes

**Quality:**

4

**Strengths And Weaknesses:**

Strengths

Solves a Critical and Practical Problem: The computational barrier to entry for training large-scale vision-language models like CLIP is a significant issue for the research community. This paper directly tackles this problem, proposing a solution that makes high-performance CLIP training more efficient and accessible.

Principled and Novel Methodology: The approach is theoretically well-grounded. Reformulating the contrastive objective as an EBM and using amortization for the partition function is a clever and powerful idea. The derivation of an efficient representation for the partition function via spectral factorization provides a solid theoretical foundation for the method, moving beyond heuristic solutions.

Strong and Comprehensive Empirical Results: The paper's claims are backed by extensive experiments. AMORLIP is shown to outperform strong baselines (CLIP, SigLIP, SogCLR, FastCLIP) across 38 diverse downstream tasks at two different scales (Table 1). The performance gains are not marginal, with relative improvements of up to 12.24% over CLIP. This demonstrates the robustness and generalizability of the learned representations.


Weaknesses

Scalability to Billion-Scale Datasets: While the experiments on CC3M and CC12M are substantial, the ultimate test for a CLIP training method is its performance at the scale of modern foundation models (e.g., LAION-5B). The authors acknowledge this limitation. While the trends strongly suggest the benefits would hold or even increase at a larger scale, demonstrating this empirically remains an important next step.

Lack of Statistical Significance Reporting: The paper does not report error bars or conduct statistical significance tests for its results. Although the consistency of improvements across 38 tasks provides strong evidence, reporting the variance across multiple training runs with different random seeds would have further strengthened the claims of superiority over baselines.

Complexity of the Framework: The proposed AMORLIP framework, with its EBM formulation, target networks, EMA updates, and specialized amortization objectives, is considerably more complex to implement than the standard CLIP InfoNCE loss. While the paper provides clear algorithmic details, this added complexity could be a minor barrier to adoption compared to the simplicity of the original CLIP.

---

> ### Author Rebuttal · Authors · 2025-07-30
>
> We appreciate the Reviewer for providing insightful comments. Please find the responses below:
>
> ## [Statistical significance reporting]
>
> >  Although the consistency of improvements across 38 tasks provides strong evidence, reporting the variance across multiple training runs with different random seeds would have further strengthened the claims of superiority over baselines.
>
> We thank the Reviewer for raising this important point. To address the concern, we conducted additional experiments with two more seeds on CC3M and report the average performance (AVG) across 38 DataComp tasks along with standard deviation (STD):
>
> | Model | AVG | STD | Improvement over CLIP |
> |------------------|---------|--------|----------------------|
> | CLIP | 21.76% | 0.30% | - |
> | FastCLIP | 23.38% | 0.25% | +1.62% |
> | AmorLIP (l2-log) | 24.07% | 0.24% | **+2.31%** |
>
> These results consistently demonstrate that AmorLIP achieves a substantial performance improvement of over 2.3% compared to CLIP, which significantly exceeds the observed variance (STD < 0.3%).
>
> ## [Complexity of the framework]
>
> >  While the paper provides clear algorithmic details, this added complexity could be a minor barrier to adoption compared to the simplicity of the original CLIP.
>
> Thank you for the feedback. We acknowledge that the proposed method introduces additional components to the conventional NCE implementation. The main complexity arises from the amortization stage, where the function $\lambda_{\theta}(\cdot)$ is optimized using the selected amortization objective. Since our proposed objectives (KL-divergence and l2-log) are inherently supported by popular ML frameworks such as PyTorch and JAX, integrating these should not pose significant difficulties. In contrast, the modifications to the contrastive learning stage are minor, as we simply replace the partition function with predictions from $\lambda_{\theta}(\cdot)$.
>
> To ensure reproducibility and ease of implementation, we have provided detailed implementation guidelines in Sec. 4 and Appendix D. Additionally, we included extensive ablation studies covering target networks, EMA updates, and amortization objectives. We hope these studies can serve as practical guidance for optimal parameter selection. Finally, we will release our implementation code, accompanied by comprehensive comments, along with the updated version of the paper to further assist our readers.
>
> ## [Comparison between two amortization objectives]
>
> >   The L2-log objective appears to perform slightly better and more consistently. Is there a theoretical or intuitive reason to prefer one over the other? For instance, are there specific data regimes or training dynamics where the unbiased but potentially higher-variance f-divergence estimator might be advantageous?
>
> Empirically, the L2-log objective demonstrates better training dynamics and stability. We provide a theoretical explanation for this observation as follows.
>
> In Appendix B3, we first show that the L2-log objective is a biased estimator of a specific instance of the f-divergence objective (i.e., KL divergence). However, as discussed in Appendix B4, the bias of the L2-log objective remains relatively small, as it closely approximates the KL divergence up to second order under mild conditions. Additionally, the L2-log formulation (Eq.10) ensures that optimization occurs entirely in log-space, effectively mitigating numerical issues associated with the exponential operation $\exp(\cdot)$ present in the KL divergence (Eq.9). Consequently, the L2-log objective reduces training variance induced by the rapidly changing temperature factor. The empirical results shown in Table 1 align well with this theoretical reasoning.
>
> In summary, we recommend using the L2-log objective when the learnable temperature undergoes significant changes during training, as commonly observed in raw CLIP pretraining (where temperature ranges broadly from approximately 14.27 to 100 according to prior efforts [2]). Conversely, in scenarios where the temperature remains stable or changes minimally (e.g., CLIP fine-tuning), the unbiased nature of the f-divergence objective could potentially offer advantages. We plan to investigate this scenario further in future studies.
>
> ## [Discussion on the EMA decay factor]
>
> >   How sensitive is the training process to the EMA decay factor α? Figure 4a shows that α=0.999 is best, but is there a "cliff" where a slightly different value leads to instability, or is the performance relatively robust within a certain range?
>
> The EMA decay factor $\alpha$ determines how much past model information is retained. Theoretically, a larger $\alpha$ stabilizes model training by smoothing gradient estimation noise and effectively preserving information from previous batches. In Figure 4(a), we clarify that the rightmost column (average accuracy across all 38 tasks) serves as the most reliable performance indicator since it aggregates results over more tasks compared to ImageNet & Shifts (5 tasks) and Retrieval (2 tasks). This overall metric clearly demonstrates a consistent trend of improved performance with increasing $\alpha$, which supports our theoretical analysis. Additionally, we observe robust performance across a wide range of $\alpha$ values, consistently surpassing baseline methods. For instance, within ImageNet & Shifts, varying $\alpha$ introduces at most a 1.1% performance difference, and even the lowest achieved accuracy (17.81% at $\alpha=0.92$) significantly exceeds the CLIP baseline (14.03%). This confirms the effectiveness and broad applicability of our stability technique across diverse hyperparameter settings.
>
> ## [Capacity of the parametric model]
>
> >   How does the capacity of this model (i.e., the MLP size) affect performance? The ablation in Table 2b suggests that smaller networks can be better, which is somewhat counter-intuitive. Could you elaborate on the hypothesis that this mitigates overfitting during the amortization stage?
>
> Thank you for raising this important point. The parametric model $\lambda_{\theta}(\cdot)$ maps a high-dimensional feature representation (e.g., 512-dimensional vector for ViT-B/32) into a positive scalar. Given this setup, the model could potentially be prone to overfitting. Reducing the model size acts as a form of capacity control and architectural regularization.
>
> A smaller network provides architectural regularization in two perspectives. First, it constrains the model to simpler functional mappings and enhances generalization. Second, it introduces minor prediction noise into the estimated partition function, effectively acting as implicit regularization for the representation learning objective (Eq. 8-10). This subtle perturbation may encourage the encoders to learn more robustly.
>
> Additionally, we emphasize that performance differences resulting from varying the model size are relatively minor (within 0.86%). This indicates the robustness of our proposed parameterization, which consistently achieves strong performance across different hyperparameter choices.
>
>
> ## Reference
>
> [1] Wei, Xiyuan, et al. "Fastclip: A suite of optimization techniques to accelerate clip training with limited resources." arXiv preprint arXiv:2407.01445 (2024).
>
> [2] Radford, Alec, et al. "Learning transferable visual models from natural language supervision." International conference on machine learning. PmLR, 2021.

---

> > ### Comment · Reviewer_31TJ · 2025-08-07
> >
> > Thank you for your rebuttal, which has addressed my concerns.

---

> > > ### Author Response · Authors · 2025-08-07
> > >
> > > Thank you for the constructive reviews and suggestions.

---

> ### Comment · Area_Chair_cb5m · 2025-08-05
>
> hi reviewer, thank you for spending time to read the paper and provide feedback. The authors have provided responses to your feedback and please spend some time to read the responses. Please do make sure let the authors know whether you still have any concerns, questions, or suggestions after reading their responses.

---

### Official Review · Reviewer_h4E3 · 2025-07-02

**Clarity:** 2
**Significance:** 2
**Originality:** 3
**Rating:** 4
**Confidence:** 4

**Summary:**

This paper introduces AmorLIP, an efficient framework for contrastive language-image pretraining that reduces reliance on large batch sizes in standard CLIP training. By amortizing partition function computations through lightweight neural networks inspired by spectral factorization, the authors propose a modular two-stage training procedure alternating between learning amortization networks and updating backbone encoders. The paper analyzes two amortization objectives, presents stability enhancements, and demonstrates improved zero-shot classification and retrieval across 38 multimodal tasks.

**Questions:**

- Could the authors discuss or report any experiments using dramatically larger pretraining datasets (e.g., LAION-400M or larger), or clarify whether the observed improvements are expected to hold in settings with hundreds of millions or billions of samples?
- Additionally, there are several tasks in Figure 2 where the improvements are negative or negligible. Could the authors analyze these cases and provide possible explanations or hypotheses about the conditions under which the proposed method may not offer significant benefits?

**Ethical Concerns:**

["NO or VERY MINOR ethics concerns only"]

**Final Justification:**

I appreciate the authors' thorough responses, which have addressed the majority of my concerns. Regarding the scaling experiments with larger datasets, I commend the authors' efforts and strongly recommend including these results in the final version. Based on the comprehensive responses and other reviewers' feedback, I will raise the rating.

**Limitations:**

Yes

**Quality:**

3

**Strengths And Weaknesses:**

**Strengths**
- The approach is technically sound and well-grounded in energy-based modeling and amortization, with clear derivations and practical reformulation.
- The method is original, leveraging spectral factorization and neural amortization, and stands out from standard buffer-based schemes.

**Weaknesses**
- All experiments are conducted with pretraining only on CC3M or CC12M, and do not empirically verify scalability to larger datasets like LAION-400M, which would strengthen the claims.
- Although the method aims to reduce dependency on large batch sizes, experiments only use batch sizes of 1k and 2k. It would be better to show how AMORLIP and CLIP perform across a wider range of batch sizes.
- SigLIP also partially addresses the large batch size issue, but Table 1 shows SigLIP underperforms CLIP at small batch sizes, which is inconsistent with previous reports. The reason for this discrepancy should be clarified.

---

> ### Author Rebuttal · Authors · 2025-07-30
>
> We thank the Reviewer for their valuable suggestion.
>
>
> ## [Experiments using larger pretraining datasets]
>
> > All experiments are conducted with pretraining only on CC3M or CC12M, and do not empirically verify scalability to larger datasets like LAION-400M, which would strengthen the claims.
>
> In our submission, we comprehensively evaluated AmorLIP and baselines using medium- (CC3M) and large-scale (CC12M) datasets across 38 tasks. As shown in Table 1 and Figure 3, AmorLIP consistently outperforms all baselines and demonstrates robust and transferable gains. Specifically, AmorLIP surpasses CLIP by 2.63% (CC3M) and 3.01% (CC12M) on average across DataComp tasks.
>
> To further investigate scalability to larger datasets, as the reviewer suggested, we have initiated additional experiments on LAION-400M with the x-large setup from [1]:
>
> | Visual Encoder | Training Epochs | Batch Size | GPUs     |
> |----------------|-----------------|------------|----------|
> | ViT-B/16       | 42 Epochs       | 640×8      | 8 × H100 |
>
> Unfortunately, despite our best efforts to secure additional compute resources, our current computational constraints mean these experiments require more than four weeks (with an average step time of ~0.9s) and cannot be completed during the rebuttal period. We appreciate the Reviewer for this suggestion and commit to finalizing these experiments and including detailed results and analyses in an updated version of our paper if permitted.
>
> ## [Ablation study on training batch size]
>
> > Although the method aims to reduce dependency on large batch sizes, experiments only use batch sizes of 1k and 2k. It would be better to show how AMORLIP and CLIP perform across  a wider range of batch sizes.
>
> We thank the Reviewer for the valuable suggestion. As our motivation was indeed to reduce dependency on large batch sizes, we initially focused on moderate batch sizes (1k and 2k). Following the Reviewer’s recommendation, we conducted additional ablation studies across a wider range of batch sizes (512 to 2048) under the medium-scale setting. We report the zero-shot IN-1K accuracy as follows:
>
> | Model | Batch size=512 | Batch size=1024 | Batch size=2048 |
> |------------------|--------|---------|---------|
> | CLIP | 15.55% | 16.84% | 18.40% |
> | FastCLIP | 19.62% | 20.58% | 21.42% |
> | **AmorLIP (l2-log)** | **20.94%** | **21.50%** | **22.07%** |
>
> These results demonstrate that AmorLIP consistently outperforms CLIP and FastCLIP[1] across varying batch sizes. Notably, AmorLIP surpasses CLIP by at least 3.68% for all batch sizes, with even greater improvements at smaller batch sizes. This further confirms the effectiveness of AmorLIP in achieving strong performance with significantly reduced batch sizes.
>
> ## [Clarification on SigLIP results]
>
> > SigLIP also partially addresses the large batch size issue, but Table 1 shows SigLIP underperforms CLIP at small batch sizes, which is inconsistent with previous reports. The reason for this discrepancy should be clarified.
>
> Thank you for raising this point. We evaluated all baseline methods, including CLIP and SigLIP, using the original OpenCLIP implementation [2], following hyperparameters from [1]. SigLIP indeed outperforms CLIP on zero-shot ImageNet&Shifts classification, aligning with observations in both the original SigLIP paper [5] and related works on similar-scale datasets (e.g., CC3M and CC12M in [3]).
>
> Despite SigLIP’s better performance on certain classification tasks, our experiments show it may underperform CLIP across the broader set of 38 DataComp tasks. This observation aligns with the theoretical analysis presented in [4]. Specifically, [4] indicates that SigLIP’s binary-NCE (classification-based) loss requires an additional input-dependent bias term for self-normalization and model consistency, which can significantly impact performance. This theoretical insight matches empirical observations from the original SigLIP paper ([5], Sec. 4.9), where it was reported that performance is sensitive to the initialization and learning dynamics of this bias term.
>
> In our experiments, we initialized SigLIP’s bias term at -10 and made it learnable throughout training, consistent with the original implementation. However, we observed that after approximately 46k training steps, SigLIP’s contrastive validation loss began to increase, indicating potential overfitting or instability under small to medium-scale settings. We hypothesize this is due to the learning complication of the bias term under small/medium scale, aligning with empirical findings from [4].
>
> In contrast, we emphasize that the proposed AmorLIP is built based on ranking-based NCE, which is explicitly self-normalized by design. This design choice eliminates the need for carefully tuning an additional bias term and thus avoids these stability issues, with relative small batch size, sharing the benefits of both CLIP and SigLip
>
> ## [Explanation on negative improvements]
>
> > Could the authors analyze these cases and provide possible explanations or hypotheses about the conditions under which the proposed method may not offer significant benefits?
>
> We observe that the proposed AmorLIP underperforms on tasks requiring recognition of highly specialized local image patterns with minimal global semantic meaning, such as PatchCamelyon and Camelyon17 (metastatic tissue classification). We hypothesize this is because these tasks emphasize repetitive local structures rather than broader semantic context.
>
> However, we emphasize that AmorLIP achieves positive gains on nearly 80% of tested tasks across both scale settings, with more than 60% of tasks showing absolute performance gains exceeding 2%. Additionally, AmorLIP demonstrates consistent improvements on widely adopted tasks, including ImageNet & Shifts and retrieval tasks on Flickr and MSCOCO, even with small batchsize, and thus, less memory cost.
>
> This highlights the effectiveness of our method. In future work, we will explore extensions of AmorLIP to better address image pattern-focused tasks, such as those encountered in PatchCamelyon.
>
> ## Reference
>
> [1] Wei, Xiyuan, et al. "Fastclip: A suite of optimization techniques to accelerate clip training with limited resources." arXiv preprint arXiv:2407.01445 (2024).
>
> [2] G. Ilharco, M. Wortsman, R. Wightman, C. Gordon, N. Carlini, R. Taori, A. Dave, V. Shankar, Namkoong, J. Miller, H. Hajishirzi, A. Farhadi, and L. Schmidt. Openclip, July 2021.
>
> [3] Wang, Bokun, et al. "On discriminative probabilistic modeling for self-supervised representation learning." arXiv preprint arXiv:2410.09156 (2024).
>
> [4] Ma, Zhuang, and Michael Collins. "Noise contrastive estimation and negative sampling for conditional models: Consistency and statistical efficiency." arXiv preprint arXiv:1809.01812 (2018).
>
> [5] Zhai, Xiaohua, et al. "Sigmoid loss for language image pre-training." Proceedings of the IEEE/CVF international conference on computer vision. 2023.

---

> > ### Comment · Reviewer_h4E3 · 2025-08-05
> >
> > I appreciate the authors' thorough responses, which have addressed the majority of my concerns. Regarding the scaling experiments with larger datasets, I commend the authors' efforts and strongly recommend including these results in the final version. Based on the comprehensive responses and other reviewers' feedback, I will raise the rating.

---

> > > ### Author Response · Authors · 2025-08-05
> > >
> > > Thank you very much for your thoughtful feedback. We will do our best to include the results of the larger-scale experiment and insights from our discussions in the final version of the paper.

---

### Comment · Area_Chair_cb5m · 2025-08-02
**Author-reviewer discussion**

Dear Authors and Reviewers,

I would like to thank the authors for providing detailed rebuttal messages.

For the **reviewers**, I would like to encourage you to carefully read all other reviews and the author responses and engage in an open exchange with the authors. Please post your first response as soon as possible within the discussion time window, so there is time for back and forth discussion with the authors. All reviewers should respond to the authors, so that the authors know their rebuttal has been read.

Cheers, AC

---

### Note · Authors · 2025-08-12

**Dear PC, SAC, AC, and Reviewers,**

Thank you for your valuable efforts and constructive feedback. We sincerely appreciate the recognition of our work’s contributions, including:

- **Principled and novel methodology** grounded in energy-based modeling and amortization (Reviewers h4E3, 31TJ, g7oi, ePbH).
- **Strong and comprehensive empirical results** across 38 downstream tasks with consistent improvements over CLIP and other strong baselines (Reviewers 31TJ, g7oi, ePbH).
- **Detailed derivations and well-structured presentation** (Reviewers h4E3, g7oi, ePbH).

During the rebuttal and discussion phases, we provided point-by-point responses addressing the majority of the reviewers’ concerns.

**Key resolutions include:**
- Conducted more comprehensive ablation studies confirming AmorLIP’s robustness and consistent gains.
- Strengthened the comparison with related works through theoretical connections and empirical analysis.
- Provided empirical guidelines for setting amortization hyperparameters.
- Presented clear discussion and explanations for the experimental results.

We have provided additional details in our individual responses and will incorporate all additions into the revised manuscript. We will also open-source our implementation for reproducibility.

We sincerely thank all reviewers, the AC, SAC, and PC for their time, effort, and constructive input, which have helped us further strengthen this work.

Best,

Authors

---

### Decision · Program_Chairs · 2025-09-17

**Decision:**

Accept (poster)

**Comment:**

This work addresses a critical and practical problem in contrastive learning, where immense computational cost and the requirement of large batch sizes for effective training.  The authors introduces AmorLIP, an efficient training framework for contrastive language-image (CLIP). AmorLIP significantly reduces the dependency on large sizes in standard CLIP training by amortizing partition function computations (the denominator in the softmax, which is the most computationally expensive part of the contrastive loss) through lightweight neural networks inspired by spectral factorization. The authors propose a modular two-stage training procedure alternating between learning amortization networks and updating backbone encoders  The authors evaluate AmorLIP on 38 zero-shot classification and retrieval tasks from the DataComp benchmark and show improved performance with AmorLIP.

Strength of the paper:
- An original method leverages spectral factorization and neural amortization. Approach is also technical sound. The use of Energy-Based Models is also an innovative idea.
- Good selection of downstream evaluation tasks provides useful understanding and support of efficacy of proposed approach.
- The work solves a critical and practical problem to reduce computational barrier to entry the area of training contrastive models like CLIP. This work proposes a solution that makes high-performance CLIP training more efficient and accessible.

Reviewers align on most of the strengths of the paper from the high level and see its novelty, contribution, and relevance to NeurIPS venue. Thus we recommend this paper to be accepted to the conference.

After most questions being addressed and clarified during the rebuttal and discussion stage, reviewers feel a few areas that can improve the paper.

To improve:
- AmorLIP was evaluated on CC3M and CC12M, while the frontier datasets used in the area has been scaled to billions (e.g., DataComp-1B, MetaCLIP). It's understandable that training with such large scale datasets could be too costly for an exploratory paper. However, we would recommend the authors 1) discussing such limitation and the gap between datasets/scale used in this work and datasets used at frontier practices, and what could be the path to close such gap in future, and 2) consider expanding experiment scale in future work. This is also applicable to the model sizes.
- Many discussion between reviewers and authors are valuable to include in the final manuscript